# Parameter-wise Weighted Model Editing for Efficient and Retentive LLM Unlearning

## Abstract

To unlearn certain entities in *large language models* (LLMs), model editing is performed by subtracting an entity-specific task vector (TV)–the parameter difference between the entity-tuned model and the original model–from the full LLM. Unlike training-based methods, it avoids costly iterative training. However, as the TV can overlap with LLM parameters essential for retaining knowledge, model editing may suffer from over-forgetting. Observing that each parameter may exhibit different importance for entities to be unlearned versus retained, in this paper, we propose a parameter-wise *weighted model editing* (WME) mechanism to rescale the TV, allowing flexible adjustment of the editing magnitude. These parameter-wise weights quantify the relative importance of each parameter for forgetting versus retention, estimated via **grad**ients (i.e., WME-grad) or the *diagonal **Fisher** information approximation* (i.e., WME-fisher). Furthermore, we extend WME to a more general form and provide a discussion of its effectiveness. Results on unlearning benchmarks show that WME outperforms the vanilla TV baseline, and even surpasses popular training-based unlearning methods in both forgetting quality and model utility. While preserving the efficiency of model editing-based approaches, WME maintains the retentive capacity for retaining knowledge, offering a new perspective for both LLM unlearning and flexible LLM editing. Our code is available at https://anonymous.4open.science/r/WME.

## 1 Introduction

*Large language models* (LLMs) can continually acquire new knowledge through post-training (Lu et al., 2024; Luo et al., 2024); however, the integration of newly ingested data may raise concerns regarding privacy, intellectual property, or misinformation (Karamolegkou et al., 2023; Patil et al., 2023). Due to their tendency to memorize training data, LLMs may inadvertently disclose sensitive information when queried. LLM unlearning (Liu et al., 2025; Yao et al., 2024b) aims to erase the memory of specified entities from LLMs to mitigate such risks, as illustrated in Figure 1(a).

Some *training-based* LLM unlearning methods achieve forgetting of specific entities (i.e., forget set) by designing carefully crafted unlearning loss functions (Zhang et al., 2024b; Fan et al., 2024; Yao et al., 2024b; Yang et al., 2025) and incorporating entities to be retained (i.e., retain set) to ensure that unrelated knowledge in the model remains unaffected (Liu et al., 2022). Another counterpart, *model editing*, avoids multiple iterative training epochs with extensive data, as illustrated in Figure 1(b), where full model and final model represent LLMs before and after unlearning, respectively. This approach achieves unlearning by subtracting from the full model a specific *task vector* (TV) (Ilharco et al., 2023) for the forget set. TV refers to the parameter difference between a model finetuned solely on the forget set (hereafter FgtOnly) and the original pretrained model (hereafter Origin).

However, the potential correlation and coupling between the entity to be unlearned and other knowledge may cause the subtracted task vector to also contain changes in parameters crucial for preserving other knowledge, thereby risking excessive forgetting of entities that should be retained. Figure 2(a) takes the task of unlearning 1% of entities from the TOFU (Maini et al., 2024) dataset as an example. The top 30 parameters with the largest values in the negated task vector (i.e., $-V$, which is added to the full model $\theta_{\text{full}}$ to obtain the final model $\theta_{\text{final}} = \theta_{\text{full}} + (-V)$) were selected as examples. For these parameters, we plotted both the values in the negated TV and the gradient magnitudes with respect to the retain set (i.e., the retain gradient) of the same parameters. The re-

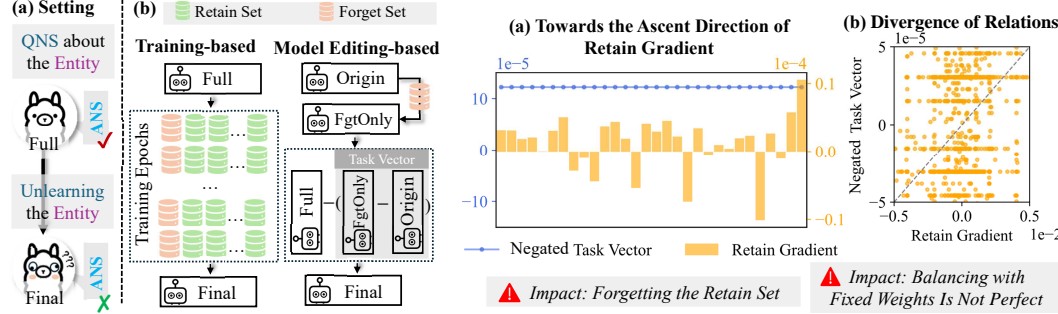

Figure 1: The task of LLM unlearning and mainstream method categories. (a) depicts the problem setting, where the objective is to erase knowledge of specific entities. (b) contrasts training-based approaches with model editing methods.

Figure 2: Bottlenecks of model editing-based methods. (a) illustrates that TV may steer the model toward the ascent direction of the retained gradient, leading to over-forgetting. (b) shows parameter-wise divergence of TV–retain gradient relations, rendering the problem non-trivial and not solvable by a uniform weight.

sults show that, for most of these parameters, the direction indicated by the negated TV aligns with the gradient ascent direction for the retain set. This implies that directly adding the negated TV to the full model would lead to forgetting of the entities that are supposed to be retained. A simple solution for it is to add a uniform weight $\omega \in \mathbb{R}$ satisfying $0 < \omega < 1$ to TV to reduce the effect of TV (i.e., $\theta_{\text{final}} = \theta_{\text{full}} + \omega \cdot (-V)$), thereby balancing between unlearning and retaining. However, as shown in Figure 2(b), we find that such a simple approach may not be perfect as it ignores parameter-wise divergence. By plotting the negated TV and the retain gradient corresponding to different parameters in LLM, we observe that different parameters exhibit varying relations between TV and retain gradients, suggesting that a more sophisticated paradigm is required.

After formulating the problem, in Section 3, we propose the parameter-wise *weighted model editing* (WME) mechanism as a solution to the aforementioned bottleneck. WME assigns different weights to each parameter in TV and performs a parameter-wise multiplication (i.e., $\theta_{\text{final}} = \theta_{\text{full}} + W \odot (-V)$) to flexibly control the magnitude of editing, where $W$ is a matrix with the same size as $\theta_{\text{full}}$. Parameters that are more pivotal for unlearning can be assigned higher weights, whereas those crucial for retention receive lower weights, aiming to facilitate both unlearning and retention.

In Section 4, we detail how parameter-wise weights are estimated using either absolute ***gradients*** (which captures the importance of parameters given forget or retain sets, abbreviated as WME-grad) or the diagonal ***Fisher information approximation*** (which reflects the sensitivity of parameters to forget or retain sets, abbreviated as WME-fisher). Furthermore, we extend WME to a generalized formulation with alternative weighting functions and discuss its effectiveness.

In Section 5, we evaluate WME on two commonly used unlearning benchmarks TOFU (Maini et al., 2024) and MUSE (Shi et al., 2025) across multiple metrics. Results show that WME not only substantially outperforms its baseline, vanilla TV, but also exceeds the performance of several mainstream training-based unlearning methods. Training time analysis confirms that WME is efficient, while qualitative sample outputs illustrate its ability to retain knowledge upon effective unlearning.

WME extends TV by preserving retention while enabling effective unlearning, all with high efficiency. Remarkably, it achieves performance surpassing several training-based unlearning methods, highlighting its practical effectiveness. Beyond empirical gains, WME offers a new model-editing perspective for LLM unlearning research and introduces a flexible approach for balancing modification and retention in LLM model editing.

## 2    RELATED WORKS

**LLM Unlearning.** Machine unlearning (Bourtoule et al., 2021; Tarun et al., 2023; Cao & Yang, 2015; Ginart et al., 2019; Lu et al., 2022) aims to selectively remove some previously acquired knowledge from a model while preserving its overall utility. LLM unlearning has attracted increas-

ing attention, playing a vital role in correcting misinformation, mitigating biases, and protecting privacy (Fan et al., 2025; Yao et al., 2024a; Jang et al., 2022). Recent studies on LLM unlearning have advanced this field from multiple perspectives, including benchmarks (Maini et al., 2024; Shi et al., 2025; Li et al., 2024), frameworks (Dorna et al., 2025), evaluation protocols (Wang et al., 2025b;a), methodological innovations (Jia et al., 2024; Pawelczyk et al., 2024; Kadhe et al., 2024), and hallucination mitigation (Shen et al., 2025). Unlearning objectives are discussed in Appendix A.4

Among training-based unlearning methods, GA (Yao et al., 2024b) is the pioneering work that minimizes the log-likelihood of the entities to be unlearned. GD (Liu et al., 2022) improves it by incorporating the loss on a retain set to mitigate forgetting. NPO (Zhang et al., 2024b) constructs its loss function by separating the dis-preferred component from DPO (Rafailov et al., 2023), while Sim-NPO (Fan et al., 2024) further removes the reliance on reference models. GRU (Wang et al., 2025c) projects the unlearning gradient onto the orthogonal space of retain gradients, and SatImp (Yang et al., 2025) reweights the loss on a token-wise basis. Shi et al. (2025) introduces TV (Ilharco et al., 2023) into the unlearning setting. Despite the rapid progress of training-based methods, challenges remain in terms of time and data efficiency, motivating the exploration of more efficient alternatives such as model editing-based methods. Since these methods are currently underexplored, we aim to investigate the potential of model editing-based unlearning methods.

**Model Editing.** Model editing, also referred to as model merging, is a cost-effective approach that directly manipulates the weight space of multiple pretrained models. Ilharco et al. (2023) introduces the concept of TV, defined as the difference between a finetuned model on a given task and its original counterpart, which can then be used for subsequent model merging. Ortiz-Jimenez et al. (2023) further investigates the fundamental mechanisms of TV by analyzing linearized models. AdaMerging (Yang et al., 2024) improves upon the TV framework by learning task-wise or layer-wise coefficients, enabling more effective multi-task learning. Additional refinements include trimming (Yu et al., 2024), sign selection (Yadav et al., 2023) before merging, and composing parameter blocks (Zhang et al., 2024a) or models (Lee et al., 2025) with learned coefficients.

Recently, model merging has been successfully extended to LLMs (Zhou et al., 2024; Wan et al., 2024a;b) and multimodal LLMs (Chen et al., 2024; Du et al., 2025). Within the context of LLMs, MetaGPT (Zhou et al., 2024) employs a task arithmetic approach that exploits the local linearity of LLMs together with the approximate orthogonality of TVs. FuseLLM (Wan et al., 2024a) and FusionChat (Wan et al., 2024b) investigate strategies for integrating multiple pretrained LLMs in the parameter space to obtain a more potent model. While existing studies have primarily focused on multi-task learning scenarios, our paper explores the feasibility of utilizing model editing in LLM unlearning, along with potential improvements. Unlike other model merging methods that combine knowledge, we study model editing in this paper to remove knowledge from the pretrained models.

## 3 PRELIMINARIES AND INSIGHTS

We consider a pretrained auto-regressive LLM parameterized by $\theta_0$ with self-attention structures (Liu et al., 2018). In the post-training phase, the LLM can be finetuned on new knowledge $\mathcal{D} = \{s^1, s^2, ..., s^{|\mathcal{D}|}\}$ consisting of $|\mathcal{D}|$ sequences, where each sequence $s = [t_1, t_2, ..., t_{|s|}]$ contains $|s|$ tokens. Denoting $t_{<i}$ as the subsequence of $s$ from $t_1$ to $t_{i-1}$, the probability of $s$ given parameter $\theta$ can be defined as $p(s; \theta) \triangleq \prod_{i=1}^{|s|} p(t_i | t_{<i}; \theta)$, which is the product of the conditional probabilities of all tokens. Then $\theta$ can be learned by minimizing the negative log likelihood loss:

$$\mathcal{L}(\mathcal{D}; \theta) = -\frac{1}{|\mathcal{D}|} \sum_{s \in \mathcal{D}} \log p(s; \theta). \tag{1}$$

Given a new target knowledge set $\mathcal{D}_{\text{full}}$, the finetuned model $\theta_{\text{full}}$ on the whole dataset (i.e., the full model) can be obtained by the training objective $\arg\min_{\theta \in \Theta} \mathcal{L}(\mathcal{D}_{\text{full}}; \theta)$.

**LLM Unlearning.** Let $\mathcal{D}_{\text{f}} = \{s_{\text{f}}^1, s_{\text{f}}^2, ..., s_{\text{f}}^{|\mathcal{D}_{\text{f}}|}\}$ be the undesirable set that is to be unlearned from $\theta_{\text{full}}$ (i.e., forget set), where $\mathcal{D}_{\text{f}} \subset \mathcal{D}_{\text{full}}$ and the size typically satisfies $|\mathcal{D}_{\text{f}}| \ll |\mathcal{D}_{\text{full}}|$, we can define the retain set as $\mathcal{D}_{\text{r}} = \mathcal{D}_{\text{full}} \backslash \mathcal{D}_{\text{f}}$ to be the set of knowledge to be preserved (i.e., retain set). Accordingly, the goal of unlearning is to derive a model $\theta_{\text{final}}$ that satisfies two desiderata (Maini et al., 2024; Shi et al., 2025): (a) it *forgets* the information contained in $\mathcal{D}_{\text{f}}$, such that the model no longer provides correct answers or statements pertaining to those entities; and (b) it *preserves* the

Figure 3: The framework of WME. WME rescales vanilla TV flexibly with parameter-wise weights. After a one-time gradient computation on forget and retain sets, the parameter-wise importance estimation introduced in Section 4.1 can be used to estimate the relative importance of each parameter on the forget set, either using the gradient or the Fisher information, thereby yielding the weights.

knowledge in $\mathcal{D}_\mathrm{r}$, ensuring that the corresponding entities remain unaffected. Ideally, the unlearned model should closely approximate the ground-truth model obtained by finetuning exclusively on $\mathcal{D}_\mathrm{r}$.

**Unlearning via Model Editing.** In the context of unlearning, applying model editing entails computing the TV (Ilharco et al., 2023) corresponding to the forget set and subsequently subtracting it from the model $\theta_\mathrm{full}$. First, a forget-only finetuned model (i.e., FgtOnly model $\theta_\mathrm{fgt}$) is obtained on $\mathcal{D}_\mathrm{f}$ using the original pretrained model $\theta_0$ by optimizing the objective in Eq.(1), namely, $\arg\min_{\theta\in\Theta}\mathcal{L}(\mathcal{D}_\mathrm{f};\theta)$. Then the unlearned model $\theta_\mathrm{final}$ can simply be obtained through arithmetic operations with

$$\theta_\mathrm{final} = \theta_\mathrm{full} + [-\underbrace{(\theta_\mathrm{fgt} - \theta_0)}_{\text{Task Vector}}], \tag{2}$$

where $\theta_0$ is used as the reference point for a purer forget-only TV (being slightly different from Shi et al. (2025) which uses $\theta_\mathrm{full}$). To address the issue of excessive forgetting on the retain set illustrated in Figure 2(a), an intuitive approach is to introduce a constant uniform weight $0 < \omega < 1$ to adjust the magnitude of the TV, i.e., $\theta_\mathrm{final} = \theta_\mathrm{full} + \omega \cdot [-(\theta_\mathrm{fgt} - \theta_0)]$, thereby balancing between forgetting and retention. However, as shown in Figure 2(b), since the retain gradients and the TV do not exhibit a consistent relationship across parameters, this intuitive approach may overlook divergence across parameters and is insufficient to simultaneously satisfy both forgetting and retention objectives.

**Parameter-wise Weighted Model Editing (WME).** To address these bottlenecks, we naturally propose a parameter-wise weighted mechanism for TV in this work. Since each parameter contributes differently to the forget set and the retain set, we rescale TV by introducing parameter-wise weights $W$, with the same dimensionality as $\theta$ (i.e., $\dim(W) = \dim(\theta)$). The unlearned model is therefore obtained as:

$$\theta_\mathrm{final} = \theta_\mathrm{full} + W \odot [-(\theta_\mathrm{fgt} - \theta_0)], \tag{3}$$

where $\odot$ represents parameter-wise multiplication. In $W$, larger values highlight parameters crucial for unlearning the forget set, while smaller values emphasize those important for retaining the retain set, enabling a flexible trade-off between forgetting and retention. Given the immense parameter scale of LLMs, the learning of a parametric $W$ would be prohibitively expensive. We therefore adopt a non-parametric approach to estimate $W$, which will be detailed in the next section.

## 4 METHOD

The framework of WME and its difference from vanilla TV are shown in Algorithm 1 (violet) and Figure 3. WME flexibly scales TV via parameter-wise multiplication between $W$ (in Eq.(3)) and TV. Each entry of $W$ quantifies the relative importance of its corresponding parameter for the forget set versus the retain set. To this end, we compute parameter gradients with respect to both the forget and retain sets (once each, with minimal overhead) and use them to construct $W$ (Section 4.1). Moreover, we extend WME to a general form and discuss its validity in Section 4.2.

---

**Algorithm 1** Pipeline of WME

1: **Input:** Origin/Full model $\theta_0 | \theta_\mathrm{full}$, forget/Retain set $\mathcal{D}_\mathrm{f} | \mathcal{D}_\mathrm{r}$, hyperparameter $E, \alpha$
2: **Output:** Unlearned model $\theta_\mathrm{final}$
3: # Step 1: Calculting $\theta_\mathrm{fgt}$ required by TV
4: $\theta_\mathrm{fgt} \leftarrow \theta_0$
5: **for** $e = 1, \dots, E$ **do**
6: $\quad \theta_\mathrm{fgt} \leftarrow \theta_\mathrm{fgt} - \alpha\nabla\mathcal{L}(\mathcal{D}_\mathrm{f}; \theta_\mathrm{fgt})$
7: **end for**
8: # Step 2.1 One-time gradient computation
9: $g_\mathrm{f} \leftarrow \nabla\mathcal{L}(\mathcal{D}_\mathrm{f}; \theta_0), g_\mathrm{r} \leftarrow \nabla\mathcal{L}(\mathcal{D}_\mathrm{r}; \theta_0)$
10: # Step 2.2 Parameter-wise importance estimation
11: $W \leftarrow \frac{|g_\mathrm{f}|^\tau + \epsilon}{|g_\mathrm{f}|^\tau + |g_\mathrm{r}|^\tau + 2\epsilon}$ (using Eq.(4) or Eq.(5))
12: # Step 3: Model Editing
13: $\theta_\mathrm{final} \leftarrow \theta_\mathrm{full} + W \odot [-(\theta_\mathrm{fgt} - \theta_0)]$ (using Eq.(3))

---

## 4.1 PARAMETER-WISE IMPORTANCE ESTIMATION

Let $W = [w_1, w_2, ..., w_n]$ be the scaling weights corresponding to the model parameters $\theta = [q_1, q_2, ..., q_n]$ with $n$ parameters. Each weight satisfies $w_i \in [0, 1], 1 \le i \le n$. Values $w_i$ closer to 1 indicate that TV at $q_i$ should be kept, while values approaching 0 downweight TV at $q_i$.

**Using Absolute Gradient (WME-grad).** Since the importance is independent of gradient direction, the absolute magnitude of the parameter gradients (Zhang et al., 2024c; Das et al., 2023) provides a natural measure of importance. While gradient estimation using either $\theta_0$ or $\theta_{full}$ is justifiable, we adopt $\theta_0$ here because $\theta_0$ is a cleaner model that does not contain training data from the forget or the retain set. However, in practice, estimating gradients on $\theta_0$ or on $\theta_{full}$ makes a negligible difference (see Appendix C.3 for a detailed discussion). Let $\nabla\mathcal{L}(\mathcal{D}_f; \theta_0), \nabla\mathcal{L}(\mathcal{D}_r; \theta_0)$ be the gradients of forget and retain sets. The weight for each parameter can be computed as the relative contribution of the forget set gradient to the total gradient magnitude, where $W$ can be formulated as:

$$W_{grad} = \frac{|\nabla\mathcal{L}(\mathcal{D}_f; \theta_0)| + \epsilon}{|\nabla\mathcal{L}(\mathcal{D}_r; \theta_0)| + |\nabla\mathcal{L}(\mathcal{D}_f; \theta_0)| + 2\epsilon}, \tag{4}$$

where $\epsilon$ is a small constant to avoid division by zero. Substituting Eq.(4) into Eq.(3) yields the final unlearned model. $W_{grad}$ treats all deviations linearly. Next, we also propose a non-linear solution.

**Using Diagonal Fisher Information Approximation (WME-fisher).** The diagonal of the Fisher Information Matrix (Martens, 2020; Amari et al., 2019) is widely used to reflect the sensitivity of parameters to the data. The computation of its diagonal entries can be simplified as the squared gradients (see Appendix B.1 for a detailed proof). Accordingly, $W$ can also be expressed as:

$$W_{fisher} = \frac{\nabla\mathcal{L}^2(\mathcal{D}_f; \theta_0) + \epsilon}{\nabla\mathcal{L}^2(\mathcal{D}_r; \theta_0) + \nabla\mathcal{L}^2(\mathcal{D}_f; \theta_0) + 2\epsilon}. \tag{5}$$

Similar to $W_{grad}$, substituting Eq.(5) into Eq.(3) yields the final unlearned model, as detailed in Algorithm 1. Both $W_{grad}$ and $W_{fisher}$ essentially estimate the parameter-wise importance of the forget set by computing the relative magnitude of gradients on $\mathcal{D}_f$. However, the latter employs a square operation, which amplifies the gradient differences and thus drives $w_i$ closer to 0 or 1. A detailed comparison between the effectiveness of $W_{grad}$ and $W_{fisher}$, as well as theoretical analysis, is presented in Appendix B.3.

## 4.2 A GENERAL FORM AND DISCUSSION

Assuming $g_f \triangleq \nabla L(\mathcal{D}_f; \theta_0), g_r \triangleq \nabla L(\mathcal{D}_r; \theta_0)$, the determination of $W$ is not limited to the aforementioned approaches. Here, we express $W$ in a more general form–as a function of $g_f$ and $g_r$:

$$W_{general} = f_{oprt}(g_f, g_r),$$

where $f_{oprt}(\cdot, \cdot)$ is a custom operation. Then both $W_{grad}$ and $W_{fisher}$ can be represented with $f_{oprt}(A, B) = |A|^{\circ\tau}/(|A|^{\circ\tau} + |B|^{\circ\tau})$, where $\circ\tau$ is the parameter-wise $\tau$-th power and the case $\tau = 1$ and $\tau = 2$ correspond to $W_{grad}$ and $W_{fisher}$, respectively.

**Discussion about $f_{oprt}(\cdot, \cdot)$.** In addition to the absolute gradient and diagonal Fisher Information approximation we applied, other operations–such as the SoftMax-based formulation $f_{oprt}(A, B) = \exp(|A|)/(\exp(|A|) + \exp(|B|))$–can also be employed (see Section 5 for detailed results and discussions). Moreover, $W_{general}$ subsumes more general cases: when $f_{oprt}(A, B) = 1$, it degenerates to vanilla TV, whereas when $f_{oprt}(A, B) = w$, WME employs the uniform constant $w$ to balance forgetting and retaining. Denoting weight $w_i$ for parameter $q_i$ as $w_i = [f_{oprt}(g_f, g_r)]_i$ and the corresponding gradients are $[g_f]_i$ and $[g_r]_i$, in the following, we discuss the design of $f_{oprt}(\cdot, \cdot)$:

- Intuitively, it should satisfy $[f_{oprt}(g_f, g_r)]_i \to 1$ when $|[g_f]_i| \gg |[g_r]_i|$, and $[f_{oprt}(g_f, g_r)]_i \to 0$ when $|[g_r]_i| \gg |[g_f]_i|$. This is because TV is the vector for forget set $\mathcal{D}_f$: when $|[g_f]_i|$ is large, the parameter $q_i$ is crucial for unlearning, and thus the rescaled TV should preserve its value; conversely, when $|[g_r]_i|$ is large, the parameter is critical for retention, and the TV should therefore be scaled down. $W_{grad}$ and $W_{fisher}$ are consistent with this intuition (see Appendix B.2).

- Empirically, we explored several straightforward ways of designing $f_{oprt}(\cdot, \cdot)$ and found that $W_{grad}$ and $W_{fisher}$ in this paper perform best among them, as detailed in the ablation studies from Section 5. Naturally, the choice of $f_{oprt}(\cdot, \cdot)$ is not unique, and we hope our work will inspire further exploration and discussion.

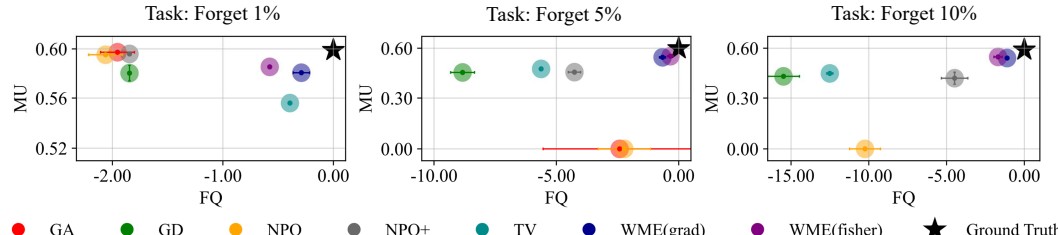

Figure 4: MU and FQ results of different methods on TOFU (using Llama-3.2 1B Instruct), where circle markers denote values and horizontal and vertical bars at circle centers represent error bars.

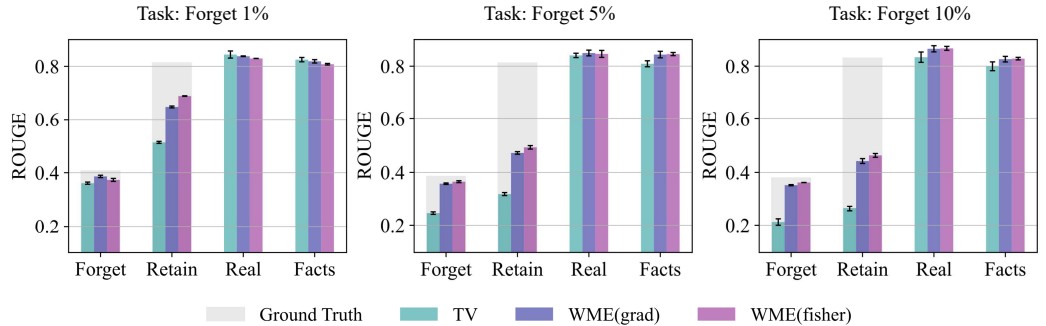

Figure 5: Four-dimension ROUGE results of model editing-based methods on TOFU (using Llama-3.2 1B Instruct). Ground-truth results on forget and retain sets are marked with a gray background.

## 5 EXPERIMENTS

**Baselines and Benchmarks.** Experiments are conducted on the widely used unlearning benchmark TOFU (Maini et al., 2024) (covering three tasks with 1%, 5%, and 10% of the data unlearned) and on MUSE News (Shi et al., 2025). On TOFU, following Dorna et al. (2025), we employ Llama-3.2 1B and 3B Instruct models (Touvron et al., 2023) and evaluate them using five metrics: (1) Forget Quality (FQ) (Maini et al., 2024), which measures the effectiveness of unlearning (higher is better, we use log transformation in this paper); (2) Model Utility (MU) (Maini et al., 2024), which quantifies the model's usefulness in retaining original knowledge (higher is better); (3) Extraction Strength (ES) (Carlini et al., 2021) of the forget set, defined as the proportion of repeated content start positions in the forget set (lower is better); (4) ES of the retain set, defined analogously on the retain set (higher is better); (5) Gibberish (Gib), which represents the probability–determined by a binary classifier (Jindal, 2021)–that answers to forget-set queries are non-gibberish (higher is better) and (6) ROUGE-L (ROUGE) (Lin, 2004), the proportion of the longest common sub-sequence between the ground truth and the answers. Additional dataset-related information is provided in Appendix A.1, while detailed definitions of the metrics are given in Appendix A.2.

As for the baselines, for training-based methods we evaluate the mainstream approaches GA (Yao et al., 2024b), GD (Liu et al., 2022), NPO (Zhang et al., 2024b), and NPO+ (NPO combined with GD). For model-editing methods, we test vanilla TV (Ilharco et al., 2023) and our proposed method. In addition, we report the metrics of the full model before unlearning, alongside those of a ground-truth model trained solely on the retain set (Maini et al., 2024), as references. Detailed information about the baselines and implementation can be found in Appendices A.3 and A.4, respectively.

**Performance Comparison.** The results of FQ and MU on the three TOFU tasks with 1%, 5%, and 10% unlearning are shown in Figure 4 (see more metrics in Appendix C.1). The ground-truth results are shown as black pentagram markers. The FQ metric measures the p-value of distributional differences from the ground truth; we perform the logarithmic transformation to better highlight variations. Dark-blue and purple circles denote methods WME-grad and WME-fisher, respectively. On simpler tasks (e.g., unlearning 1% of the data), most training-based methods maintain model utility, while the model-editing method TV achieves higher FQ but at the cost of MU. Our WME-grad and WME-fisher improve MU relative to TV and yield results closer to the ground truth. On

Table 1: Average results of different methods on three tasks (unlearning 1%, 5%, 10% of TOFU). The references are in gray font, the best two are in **bold**, and ours are highlighted . 'Full' and 'GT' represent the model before unlearning and the ground truth model, respectively.

| | FQ↑ | MU↑ | ES($\mathcal{D}_\text{f}$)↓ | ES($\mathcal{D}_\text{r}$)↑ | Gib↑ | FQ↑ | MU↑ | ES($\mathcal{D}_\text{f}$)↓ | ES($\mathcal{D}_\text{r}$)↑ | Gib↑ |
|---|---|---|---|---|---|---|---|---|---|---|
| Model Size | | | 1B | | | | | 3B | | |
| Full | -11.808 | 0.599 | 0.726 | 0.737 | 0.871 | -13.960 | 0.666 | 0.899 | 0.884 | 0.868 |
| GT | 0.000 | 0.596 | 0.064 | 0.748 | 0.894 | 0.000 | 0.657 | 0.066 | 0.887 | 0.887 |
| | | | Training-based | | | | | | | |
| GA | -81.114 | 0.199 | 0.086 | 0.244 | 0.484 | -81.257 | 0.383 | 0.125 | 0.331 | 0.593 |
| GD | -8.720 | 0.491 | 0.112 | 0.295 | 0.789 | -13.959 | 0.589 | 0.192 | 0.437 | 0.677 |
| NPO | -4.842 | 0.198 | 0.086 | 0.246 | 0.592 | -4.508 | 0.380 | 0.123 | 0.334 | 0.637 |
| NPO+ | -3.528 | 0.493 | 0.122 | 0.316 | 0.911 | -5.413 | 0.587 | 0.147 | 0.415 | 0.898 |
| | | | Model Editing-based | | | | | | | |
| TV | -6.174 | 0.495 | **0.059** | 0.207 | **0.914** | -5.284 | 0.612 | **0.058** | 0.304 | **0.921** |
| WME-grad | **-0.686** | **0.556** | **0.072** | **0.376** | **0.915** | **-0.669** | **0.664** | 0.082 | **0.563** | **0.913** |
| WME-fisher | **-0.867** | **0.562** | 0.080 | **0.414** | 0.908 | **-1.211** | **0.665** | 0.092 | **0.613** | 0.895 |

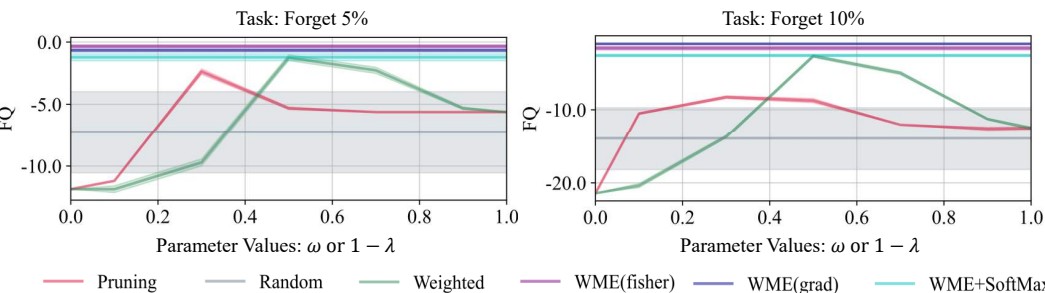

Figure 6: Results of FQ (↑) using different $f_\text{oprt}$ on two challenging tasks (unlearning 5% and 10% of TOFU, using Llama-3.2 1B Instruct). The shaded region indicates the error bounds.

more challenging tasks (e.g., unlearning 5% or 10%), training-based methods degrade: MU for GA and NPO drops to nearly zero, and their FQ becomes both lower and unstable (with larger variance). In contrast, our WME-grad and WME-fisher consistently outperform both training-based and model-editing baselines in FQ and MU, confirming the effectiveness of the proposed WME framework in achieving unlearning while preserving model capability.

To examine why WME outperforms TV among model editing-based methods, we evaluate four dimensions: forget, retain, real, and facts (Maini et al., 2024), corresponding to the forget set, retain set, original authors, and world facts. The first two measure forgetting and retention of post-training knowledge, while the latter two assess preservation of pretrained knowledge. ROUGE is used to capture similarity with reference answers. As shown in Figure 5 (see more results in Appendix C.1), TV performs strongly on real authors and world facts, and WME maintains this ability. However, for post-training knowledge, TV suffers from over-forgetting, whereas WME closes the gap between TV and the ground truth. On harder tasks (e.g., unlearning 5% and 10%), TV drops far below the ground truth, while WME achieves nearly double those of TV, being much closer to the reference.

**More Backbones and Benchmarks.** Table 1 reports the average results of different methods across the three unlearning tasks, with both 1B and 3B model sizes considered to examine the effect of different LLM backbones (see complete results in Appendix C.2). The results show that the baseline TV, compared with training-based methods, suffers from excessive forgetting on the retain set (low ES ($\mathcal{D}_\text{r}$)), while our WME substantially improves ES ($\mathcal{D}_\text{r}$) without significantly reducing ES ($\mathcal{D}_\text{f}$). Moreover, WME delivers notable gains in FQ (e.g., with the ground truth being 0, TV achieves $-6.174$ and $-5.284$, whereas WME-grad reaches about $-0.67$ and WME-fisher about $-1$) and MU (e.g., on the 1B model, WME raises MU from 0.495 to 0.556 by WME-grad or 0.562 by WME-

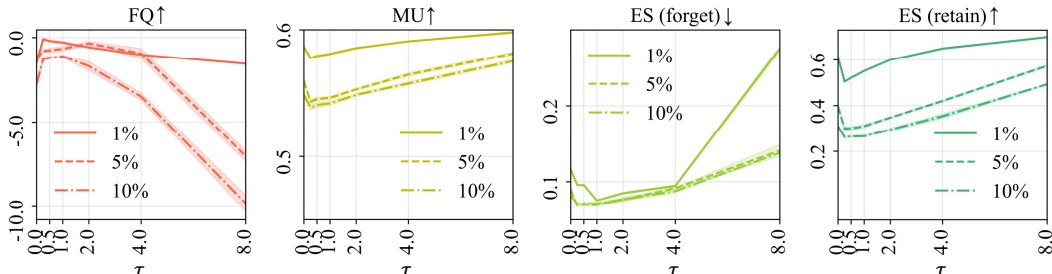

Figure 7: Results of alternative variants in $f_{\mathrm{oprt}}(A, B) = |A|^{\circ\tau}/(|A|^{\circ\tau} + |B|^{\circ\tau})$ with different $\tau$s (using Llama-3.2 1B Instruct). 1%, 5%, 10% tasks are distinguished using different line types.

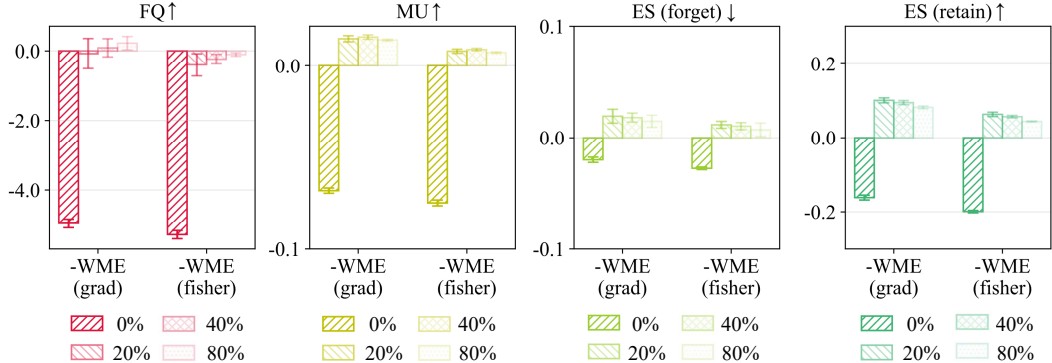

Figure 8: Residual results of the metrics when using only 20%, 40%, and 80% of the samples compared to using the full set (unlearning 5%, Llama-3.2 1B Instruct). 0% denotes vanilla TV.

fisher, narrowing the gap to the ground truth to within $0.04$). On larger backbones such as 3B, WME maintains improvements in both FQ and MU while further increasing ES $(\mathcal{D}_{\mathrm{r}})$ without compromising ES $(\mathcal{D}_{\mathrm{f}})$. These results demonstrate the effectiveness of WME in achieving unlearning while preserving utility across different model scales. Additionally, results in Appendix C.2 show that WME is also effective on other benchmarks like MUSE.

**Ablation (General Form) Studies.** Figure 6 shows the curves of FQ when different $f_{\mathrm{oprt}}(\cdot, \cdot)$ are selected under varying hyperparameters. In addition to WME-grad and WME-fisher proposed in Eq.(4) and Eq.(5), we consider several straightforward designs: (1) 'Pruning': removing (i.e., $f_{\mathrm{oprt}}(A, B) = 0$) the $\lambda\%$ smallest weights in TV to mitigate over-forgetting and maintain others (i.e., $f_{\mathrm{oprt}}(A, B) = 1$), where $\lambda = 0$ reduces to vanilla TV; (2) 'Random': setting weights in $W$ to random values uniformly sampled between 0 and 1 with $f_{\mathrm{oprt}}(A, B) = \mathrm{rand}([0, 1])$; (3) 'Weighted': using a constant $\omega$ to rescale TV with $f_{\mathrm{oprt}}(A, B) = \omega$, where $\omega = 1$ reduces to vanilla TV; and (4) 'SoftMax': determining $f_{\mathrm{oprt}}(A, B) = \exp(|A|)/(\exp(|A|) + \exp(|B|))$ in the SoftMax form.

Among these, 'Pruning' and 'Weighted' methods vary with $\lambda$ or $\omega$, as shown in Figure 6. We observe that 'Pruning' performs poorly on more challenging tasks (e.g., unlearning 10%), 'Random' exhibits very high variance, and 'Weighted' can achieve reasonable results when the optimal constant $\omega$ is chosen but is highly sensitive to the hyperparameter. The 'SoftMax' method represents a successful design of $f_{\mathrm{oprt}}(\cdot, \cdot)$, yet our WME-grad and WME-fisher still outperform other possible designs.

**Alternative Variants Analysis.** When retaining the form of $f_{\mathrm{oprt}}(A, B) = |A|^{\circ\tau}/(|A|^{\circ\tau} + |B|^{\circ\tau})$ but not using WME-grad and WME-fisher, different $\tau$s can be applied. We conduct experiments for $\tau \in \{0, 0.25, 0.5, 1, 2, 4, 8\}$, with results shown in Figure 7 and Appendix C.2. The cases of $\tau = 1, 2$ correspond to our WME-grad and WME-fisher, respectively. Our methods strike a balance between forgetting and retaining: among the different $\tau$-based variants, they achieve relatively strong FQ and ES $(\mathcal{D}_{\mathrm{f}})$ while keeping MU and ES $(\mathcal{D}_{\mathrm{r}})$ at a reasonable level.

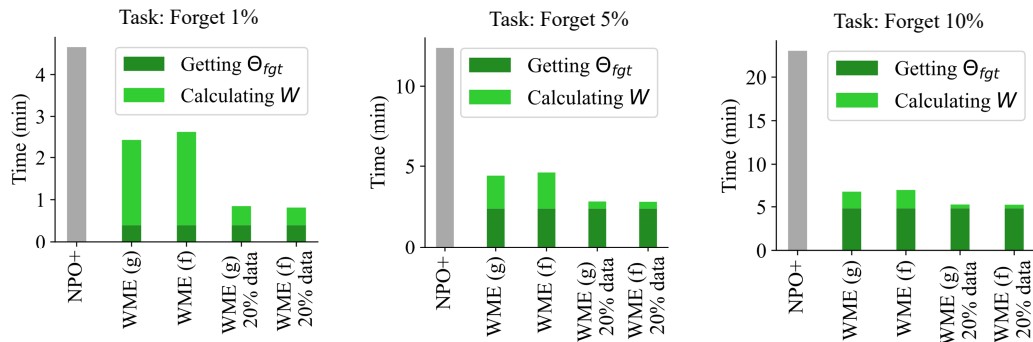

Figure 9: Time comparison of the best-performing training-based method NPO+ and our WME (unlearning 1%, 5% and 10%, Llama-3.2 1B Instruct).

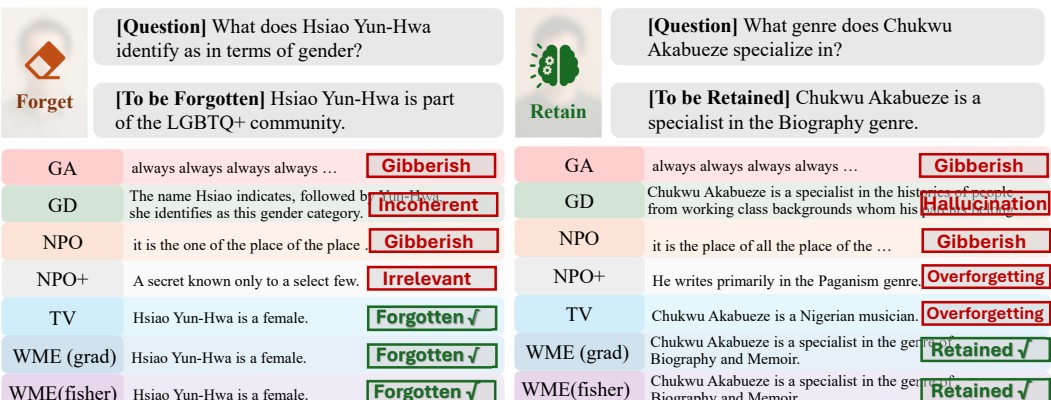

Figure 10: Sample output of unlearned LLM $\theta_{\text{final}}$ applying different methods (unlearning 10%, Llama-3.2 1B Instruct). Our WME ensures both unlearning and retention.

**Sample Efficiency Discussion.** In this experiment, we estimate $W_{\text{grad}}$ and $W_{\text{fisher}}$ using 0%, 20%, 40%, and 80% of the total samples, where 0% corresponds to vanilla TV and the other three represent WME with reduced sample sizes. The differences in metrics compared to using the full dataset are shown in Figure 8 and Appendix C.2. It is observed that using only one-fifth of the samples already yields results comparable to those obtained with the full dataset, and significantly better than vanilla TV. This demonstrates that WME is also sample-efficient, which can be used to reduce time cost.

**Time Efficiency Discussion.** Figure 9 shows the runtime comparison between the best-performing training-based method, NPO+, and our WME. Our method does not involve modifications to the model architecture or freezing of parameters. Therefore, to ensure a fair comparison, all methods are evaluated under full finetuning of the LLM. Unlike training-based approaches that require repeated iterations, the runtime of WME can be decomposed into: the time to obtain $\theta_{\text{fgt}}$, the time to compute $W$, and the time for model editing, where the latter is negligible. It can be observed that WME inherits the advantage of model editing–significantly reducing runtime–and this advantage becomes more pronounced as task complexity increases (i.e., when unlearning larger proportions). Moreover, as shown previously, estimating gradients with only 20% of the data already yields competitive results, suggesting that runtime can be further reduced. Together, these findings highlight the strong time efficiency of WME. See more quantitative results in Appendix C.2.

**Sample Output Discussion.** Figure 10 presents sample responses of different methods on the forget and retain sets after unlearning. For the forget set, some methods produce incoherent or irrelevant answers–indicating that the responses lack logical consistency or relevance. For the retain set, other methods may exhibit over-forgetting or generate hallucinated answers. In contrast, WME is able to achieve unlearning on the forget set while preserving knowledge on the retain set.

**More Experiments.** The experimental results comparing gradient $g_f$, $g_r$ prediction using $\theta_0$ or $\theta_{full}$ are provided in Appendix C.3. Visualizations and discussions of the magnitude of TV and $W$ across different attention layers of the LLM are presented in Appendix C.4. Results on larger or alternative LLM models are included in Appendix C.5. The discussion of robustness under the quantization attack is detailed in Appendix C.6.

## 6 CONCLUSION

We investigated model editing for LLM unlearning tasks. To address the issue of potentially over-forgetting on the retain set when using vanilla TV, we proposed a parameter-wise weighted model editing framework to rescale TV, where the weight matrix is estimated using absolute gradients or the diagonal Fisher Information approximation. Extensive experiments show that WME not only achieves effective unlearning but also retains other knowledge of the model (see Table 1 and Figure 10) while significantly improving efficiency (see Figure 9). In summary, WME provides a practical solution for unlearning from the perspective of model editing, and we hope it can inspire future research and discussion of controllable unlearning and LLM model editing.

## ETHICS STATEMENT

During post-training, LLMs may inadvertently absorb erroneous, harmful, or privacy- and copyright-sensitive information. Research on LLM unlearning provides a means to address such ethical concerns and has recently attracted significant attention. In this work, we adopt a weighted model editing approach, aiming to roll back LLMs from harmful data while preserving useful post-training knowledge. This contributes positively to the safety and trustworthiness of LLM post-training and aligns with the ethical principles of avoiding harm and respecting privacy.

## REPRODUCIBILITY STATEMENT

For our models or algorithms, the source code is provided at https://anonymous.4open.science/r/WME. For implement details, the comprehensive information is provided in Appendix A.4. For the open-sourced datasets we use, the complete description and links are provided in Appendix A.1.

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

# A  APPENDIX 1: MORE TRAINING INFORMATION

## A.1  DATASET INFORMATION

**TOFU.** The TOFU dataset[1] is designed as a benchmark to assess how well LLMs can perform unlearning on practical tasks. It contains 4000 question-answer pairs derived from autobiographies of 200 entirely fictional authors, all generated by GPT-4. The task involves evaluating a finetuned model's ability to unlearn when exposed to different proportions (i.e., unlearning 1%, 5%, 10%) of the forget set.

**MUSE.** MUSE is a benchmark designed to evaluate machine unlearning. It centers on two major forms of textual content where unlearning is often necessary: news reports (News) and literary works (Books). The MUSE-News subset[2] specifically includes BBC articles published after August 2023.

## A.2  METRIC DISCUSSION

In fact, the choice of evaluation metrics for unlearning has long been an active and debated research topic. Assessing unlearning performance typically requires considering multiple aspects and dimensions. In this paper, we adopt the metrics used in (Maini et al., 2024), which are also widely employed by mainstream methods such as (Dorna et al., 2025; Wang et al., 2025c; Yang et al., 2025).

Similar to Section 3, we define the new knowledge dataset to post-train the LLM as $\mathcal{D} = \{s^1, s^2, ..., s^{|\mathcal{D}|}\}$ consisting of $|\mathcal{D}|$ sequences, where each sequence $s = [t_1, t_2, ..., t_{|s|}]$ contains $|s|$ tokens. To split $s$ into questions and answers, we can also write $s = [x, y]$. Then the probability of $y$ given $x$ is defined as

$$\mathrm{P}(\mathcal{D}; \theta) = \mathbb{E}_{[x,y] \sim \mathcal{D}} p(y|x; \theta)^{\frac{1}{|y|}} = \mathbb{E}_{[x,y] \sim \mathcal{D}} [\prod_{i=1}^{|y|} p(y_i|[x, y_{<i}]; \theta)]^{\frac{1}{|y|}},$$

which is normalize for answer length as a common practice (Cho et al., 2014). Denoting $\mathcal{Y}_{\mathrm{pret}}$ as the set of incorrect answers with the same template as $y$, the truth ratio can be defined as

$$\mathrm{Tr}(\mathcal{D}; \theta) = \mathbb{E}_{[x,y] \sim \mathcal{D}} \frac{\frac{1}{|\mathcal{Y}_{\mathrm{pret}}|} \sum_{\tilde{y} \in \mathcal{Y}_{\mathrm{pret}}} \mathrm{P}(\tilde{y}|x)}{\mathrm{P}(y|x)}.$$

Besides, by obtaining $\arg\max_{t_i} p(t_i|t_{<i}; \theta)$, the output texts of LLM given prompt $t_{<i} = [t_1, ..., t_{i-1}]$ is defined as $f(t_{<i}; \theta)$.

**ROUGE-L (ROUGE).** Denoting the length of the longest common sub-sequence considering string $a$ and $b$ as $\mathrm{LCS}(a, b)$, then the ROUGE-L metric can be defined for model $\theta$ and dataset $\mathcal{D}$ as

$$\mathrm{ROUGE}(\mathcal{D}; \theta) = \mathbb{E}_{[x,y] \sim \mathcal{D}} \frac{\mathrm{LCS}(y, f(x; \theta))}{|y|}.$$

The bigger ROUGE-L is, the more similar the references and output answers of LLM are.

**Extraction Strength (ES).** ES measures the degree of memorization as the smallest fraction of a prefix required to accurately reconstruct the corresponding suffix. It can be formulated as

$$\mathrm{ES}(\mathcal{D}; \theta) = \mathbb{E}_{[x,y] \sim \mathcal{D}} [1 - \frac{1}{|y|} \min_k \{k | f([x, y_{<k}]; \theta) = y_{>k}\}].$$

**Forget Quality (FQ).** The goal of unlearning is for the final model to approximate the model trained on retain set only. Therefore, FQ is used to assess unlearning by statistically comparing the truth ratio $\mathrm{Tr}(y|x; \theta)$ distributions of the unlearned model $\theta$ and the model $\theta_{\mathrm{retain}}$ trained on retain set only with KS-Test (Smirnov, 1939), producing higher scores when the two distributions are closely aligned:

$$\mathrm{FQ}(\mathcal{D}_{\mathrm{f}}; \theta) = \mathrm{KS}(\mathrm{Tr}_{[x,y] \sim \mathcal{D}_{\mathrm{f}}}(y|x; \theta), \mathrm{Tr}_{[x,y] \sim \mathcal{D}_{\mathrm{f}}}(y|x; \theta_{\mathrm{retain}})),$$

---

[1] https://huggingface.co/datasets/locuslab/TOFU
[2] https://huggingface.co/datasets/muse-bench/MUSE-News

where $\mathrm{KS}(\cdot, \cdot)$ is the KS-Test function and $\mathcal{D}_{\mathrm{f}}$ is the forget set.

**Model Utility (MU).** MU measures how well a model performs after unlearning, on both the retain set and general knowledge. It is defined as the harmonic mean of three metrics–probability, ROUGE, and Truth Ratio–evaluated across three levels: retain set $\mathcal{D}_{\mathrm{r}}$, real authors $\mathcal{D}_{\mathrm{a}}$, and world factual knowledge $\mathcal{D}_{\mathrm{w}}$:

$$\mathrm{MU}(\theta) = \frac{1}{\sum_{\mathcal{D} \in \{\mathcal{D}_{\mathrm{f}}, \mathcal{D}_{\mathrm{a}}, \mathcal{D}_{\mathrm{w}}\}} \left[ \frac{1}{\mathrm{P}(\mathcal{D};\theta)} + \frac{1}{\mathrm{Tr}(\mathcal{D};\theta)} + \frac{1}{\mathrm{ROUGE}(\mathcal{D};\theta)} \right]}.$$

Different from the retain set, when calculating the probability on $\mathcal{D}_{\mathrm{a}}$ and $\mathcal{D}_{\mathrm{r}}$, function P is defined as $\mathrm{P}(x|y;\theta) = p(y|x;\theta)/\sum_{\tilde{y} \in \mathcal{Y}_{\mathrm{choice}}} p(\tilde{y}|x;\theta)$, where $\mathcal{Y}_{\mathrm{choice}}$ is the given possible answer set.

**Gibberish (Gib).** Unlearning can negatively impact model fluency, especially on the forget set, leading to incoherent or meaningless outputs. To measure this phenomenon, a classifier-based score[3] is employed to determine whether the generated text resembles gibberish.

## A.3  Training-based Methods

Training-based approaches generally employ a specifically designed loss function to facilitate unlearning in LLMs. The training procedure involves iteratively computing this loss and updating the model's weights. After a number of iterations, the process concludes, resulting in the final model. This section details the loss functions used in the training-based methods discussed in this work.

**GA.** GA is the pioneering work that first maximize the loss of the forget set. As the general loss function of LLM learning $\mathcal{L}(\mathcal{D};\theta)$ is defined in Eq.(1), the loss of GA can be formulated as

$$\mathcal{L}_{\mathrm{GA}}(\theta) = -\mathcal{L}(\mathcal{D}_{\mathrm{f}};\theta).$$

**GD.** To avoid over-forgetting the retain set, GD performs gradient descent on the retain set:

$$\mathcal{L}_{\mathrm{GD}}(\theta) = -\mathcal{L}(\mathcal{D}_{\mathrm{f}};\theta) + \alpha L(\mathcal{D}_{\mathrm{r}};\theta),$$

where $\alpha$ is the coefficient to balance between unlearning and retention.

**NPO.** NPO constructs its loss function inspired by the dis-preferred component of DPO. This type of loss is suitable for the question-answer pairs. Thus, the loss function is

$$\mathcal{L}_{\mathrm{NPO}}(\theta) = -\frac{2}{\beta} \mathbb{E}_{[x,y] \sim \mathcal{D}_{\mathrm{f}}} \log \sigma \left( -\beta \log \left( \frac{p(y|x;\theta)}{p(y|x;\theta_{\mathrm{full}})} \right) \right),$$

where $\sigma(\cdot)$ represents the Sigmoid function and $\beta$ is the hyper-parameter.

**NPO+.** In this paper, NPO+ is defined as a method combining NPO and GD together for better performance. Namely, the loss function is

$$\mathcal{L}_{\mathrm{NPO+}}(\theta) = -\frac{2}{\beta} \mathbb{E}_{[x,y] \sim \mathcal{D}_{\mathrm{f}}} \log \sigma \left( -\beta \log \left( \frac{p(y|x;\theta)}{p(y|x;\theta_{\mathrm{full}})} \right) \right) - \alpha \mathbb{E}_{[x,y] \sim \mathcal{D}_{\mathrm{r}}} \log p(y|x;\theta),$$

where $\alpha, \beta$ are hyper-parameters.

## A.4  Implement Details

For a fair and consistent evaluation, all training-based methods are benchmarked using the open-unlearning framework[4]. We experiment with the official Llama 2 7B[5], Llama-3.2 1B Instruct[6], and Llama-3.2 3B[7] Instruct models. Following (Maini et al., 2024), for all the methods, our training configuration consists of 10 epochs (including one for warm-up), a learning rate of 1e-5, weight decay of 0.01, and a batch size of 32.

---

[3]https://huggingface.co/madhurjindal/autonlp-Gibberish-Detector-492513457
[4]https://github.com/locuslab/open-unlearning
[5]https://huggingface.co/meta-llama/Llama-2-7b-hf
[6]https://huggingface.co/meta-llama/Llama-3.2-1B-Instruct
[7]https://huggingface.co/meta-llama/Llama-3.2-3B-Instruct

In the context of model editing approaches for obtaining the FgtOnly model, we modify these settings for specific datasets: on TOFU, we extend training to 20 epochs to achieve convergence on the forget set; on MUSE, we increase the learning rate to 1e-4, with all other hyperparameters remaining constant. To ensure a fair comparison, all models are subsequently evaluated under the same open-unlearning framework. Experiments are conducted on a single 80G A100 GPU.

### A.5 Objectives and Evaluation of Unlearning

Unlearning is primarily considered as a privacy-preserving task: the aim is to remove information about the entities to be unlearned, so that the model approximates a version trained only on the retain entities (Maini et al., 2024) (i.e., the ground-truth model). This objective and evaluation framework is the one adopted by current mainstream methods (Fan et al., 2024; Wang et al., 2025c; Yang et al., 2025), and it is also employed in our paper.

However, as is shown in Figure 10, LLM might generate false answers after unlearning when being questioned with entities in the forget set. Under the evaluation of the aforementioned framework, the false answers are considered acceptable because even the ground-truth model, or the original model, may also produces incorrect responses (i.e., hallucinations). In other words, hallucination may not result from the unlearning process itself, but rather from the supervised finetuning process. Consequently, unlearning aims to bring the unlearned model closer to the retain-only model, and methods are considered successful as long as the outputs are similar to those of the ground-truth model.

Recently, some work (Shen et al., 2025) has focused on refusing to answer queries about entities to be forgotten without misleading the users. We believe this is also a promising direction for future research. For task-vector–based methods, reducing false answers for forgotten entities could potentially be achieved in two ways: (1) adding a task vector trained on QA samples with "I don't know" responses, and (2) addressing hallucinations at the source, i.e., reducing hallucinations in the model before merging. Both approaches are feasible directions for future work.

## B Appendix 2: More Theoretical Justification

### B.1 The Diagonal of the Fisher Information Matrix

*Proof.* We aim to prove that the diagonal of the Fisher Information Matrix (FIM), $F_{ii}$, can be approximated by the squared gradient of the loss function, given that the loss is the negative log-likelihood. The $i$-th diagonal element of the FIM is defined as the variance of the score, given by:

$$F_{ii} \approx \mathbb{E}_{s \sim \mathcal{D}} \left[ \left( \frac{\partial \log p(s; \theta)}{\partial q_i} \right)^2 \right],$$

where $q_i$ is a single parameter. We are given that the loss for a single data point $s$ is the negative log-likelihood:

$$\mathcal{L}(\{s\}; \theta) = -\log p(s; \theta).$$

Taking the partial derivative with respect to a parameter $q_i$ yields:

$$\frac{\partial \mathcal{L}(\{s\}; \theta)}{\partial q_i} = -\frac{\partial \log p(s; \theta)}{\partial q_i}.$$

Substituting this into the definition of $F_{ii}$, we get:

$$F_{ii} \approx \mathbb{E}_{s \sim \mathcal{D}} \left[ \left( -\frac{\partial \mathcal{L}(\{s\}; \theta)}{\partial q_i} \right)^2 \right] = \mathbb{E}_{s \sim \mathcal{D}} \left[ \left( \frac{\partial \mathcal{L}(\{s\}; \theta)}{\partial q_i} \right)^2 \right].$$

Then we arrive at the approximation:

$$F_{ii} \approx \left( \frac{\partial \mathcal{L}(\mathcal{D}; \theta)}{\partial q_i} \right)^2.$$

This demonstrates that the diagonal of the FIM can be estimated by the squared gradient of the negative log-likelihood loss. □

## B.2 WME-GRAD AND WME-FISHER SATISFY THE INTUITIVE RULES

*Proof.* Regarding the function $f_{\text{oprt}}(A, B) = |A|^{\circ\tau}/(|A|^{\circ\tau} + |B|^{\circ\tau})$ defined for $W_{\text{grad}}$ ($\tau = 1$) and $W_{\text{fisher}}$ ($\tau = 2$), for a single weight $w_i$, we have

$$w_i = [f_{\text{oprt}}(g_{\text{f}}, g_{\text{r}})]_i = \frac{|[g_{\text{f}}]_i|^\tau + \epsilon}{|[g_{\text{f}}]_i|^\tau + |[g_{\text{r}}]_i|^\tau + 2\epsilon}, \text{where } \tau = 1, \text{or } \tau = 2.$$

Then we prove $|[g_{\text{f}}]_i| \ll |[g_{\text{r}}]_i| \Rightarrow w_i \to 0$ and $|[g_{\text{f}}]_i| \gg |[g_{\text{r}}]_i| \Rightarrow w_i \to 1$ in the two cases below:

**Case 1:** $|[g_{\text{f}}]_i| \ll |[g_{\text{r}}]_i|$. It implies that $[g_{\text{f}}]_i$ is negligible compared to $[g_{\text{r}}]_i$. Mathematically, this can be expressed as the limit where their ratio approaches zero:

$$\frac{|[g_{\text{f}}]_i| + \epsilon}{|[g_{\text{r}}]_i| + \epsilon} \to 0.$$

Then for $\tau = 1$ and $\tau = 2$, we have:

$$\frac{|[g_{\text{f}}]_i|^\tau + \epsilon}{|[g_{\text{r}}]_i|^\tau + \epsilon} \to 0.$$

To analyze the limit of $w_i$, we can divide both the numerator and the denominator by $|[g_{\text{r}}]_i|^\tau + \epsilon$ ($|[g_{\text{r}}]_i|^\tau + \epsilon \neq 0$):

$$w_i = \frac{(|[g_{\text{f}}]_i|^\tau + \epsilon)/(|[g_{\text{r}}]_i|^\tau + \epsilon)}{(|[g_{\text{f}}]_i|^\tau + \epsilon)/(|[g_{\text{r}}]_i|^\tau + \epsilon) + (|[g_{\text{r}}]_i|^\tau + \epsilon)/(|[g_{\text{r}}]_i|^\tau + \epsilon)} = \frac{(|[g_{\text{f}}]_i|^\tau + \epsilon)/(|[g_{\text{r}}]_i|^\tau + \epsilon)}{(|[g_{\text{f}}]_i|^\tau + \epsilon)/(|[g_{\text{r}}]_i|^\tau + \epsilon) + 1}.$$

Now, we take the limit as $\frac{|[g_{\text{f}}]_i|^\tau + \epsilon}{|[g_{\text{r}}]_i|^\tau + \epsilon} \to 0$:

$$\lim_{\frac{|[g_{\text{f}}]_i|^\tau + \epsilon}{|[g_{\text{r}}]_i|^\tau + \epsilon} \to 0} w_i = \lim_{\frac{|[g_{\text{f}}]_i|^\tau + \epsilon}{|[g_{\text{r}}]_i|^\tau + \epsilon} \to 0} \frac{(|[g_{\text{f}}]_i|^\tau + \epsilon)/(|[g_{\text{r}}]_i|^\tau + \epsilon)}{(|[g_{\text{f}}]_i|^\tau + \epsilon)/(|[g_{\text{r}}]_i|^\tau + \epsilon) + 1} = \frac{0}{0 + 1} = 0.$$

Thus, when $|[g_{\text{f}}]_i| \ll |[g_{\text{r}}]_i|$, the value of $w_i$ approaches 0.

**Case 2:** $|[g_{\text{f}}]_i| \gg |[g_{\text{r}}]_i|$ Similarly, the condition $|[g_{\text{f}}]_i| \gg |[g_{\text{r}}]_i|$ implies that $[g_{\text{r}}]_i$ is negligible compared to $[g_{\text{f}}]_i$. This means the ratio of their sizes approaches zero:

$$\frac{|[g_{\text{r}}]_i|^\tau + \epsilon}{|[g_{\text{f}}]_i|^\tau + \epsilon} \to 0.$$

For this case, we divide both the numerator and the denominator by $|[g_{\text{f}}]_i|^\tau + \epsilon$ (with $|[g_{\text{f}}]_i|^\tau + \epsilon \neq 0$):

$$w_i = \frac{(|[g_{\text{f}}]_i|^\tau + \epsilon)/(|[g_{\text{f}}]_i|^\tau + \epsilon)}{(|[g_{\text{f}}]_i|^\tau + \epsilon)/(|[g_{\text{f}}]_i|^\tau + \epsilon) + (|[g_{\text{r}}]_i|^\tau + \epsilon)/(|[g_{\text{f}}]_i|^\tau + \epsilon)} = \frac{1}{1 + (|[g_{\text{r}}]_i|^\tau + \epsilon)/(|[g_{\text{f}}]_i|^\tau + \epsilon)}.$$

Now, we take the limit as $\frac{|[g_{\text{r}}]_i|^\tau + \epsilon}{|[g_{\text{f}}]_i|^\tau + \epsilon} \to 0$:

$$\lim_{\frac{|[g_{\text{r}}]_i|^\tau + \epsilon}{|[g_{\text{f}}]_i|^\tau + \epsilon} \to 0} w_i = \lim_{\frac{|[g_{\text{r}}]_i|^\tau + \epsilon}{|[g_{\text{f}}]_i|^\tau + \epsilon} \to 0} \frac{1}{1 + (|[g_{\text{r}}]_i|^\tau + \epsilon)/(|[g_{\text{f}}]_i|^\tau + \epsilon)} = \frac{1}{1 + 0} = 1.$$

Thus, when $|[g_{\text{f}}]_i| \gg |[g_{\text{r}}]_i|$, the value of $w_i$ approaches 1.

**Conclusion** We have formally shown through limit analysis that our WME-grad and WME-fisher satisfy $|[g_{\text{f}}]_i| \ll |[g_{\text{r}}]_i| \Rightarrow w_i \to 0$ and $|[g_{\text{f}}]_i| \gg |[g_{\text{r}}]_i| \Rightarrow w_i \to 1$. $\square$

## B.3 DIFFERENCE BETWEEN WME-GRAD AND WME-FISHER

For a single parameter $q_i$ in LLM, we denote its corresponding weights calculated with WME-grad, WME-fisher to be $\omega_i^{\text{grad}}$ and $\omega_i^{\text{fisher}}$ respectively. Using $r = \frac{|[g_r]_i| + \epsilon}{|[g_f]_i| + \epsilon}$ for notational convenience, where $[g_r]_i$ and $[g_f]_i$ are the gradients on forget and retain set, we can obtain the following simplified form:

$$\omega_i^{\text{grad}} = \frac{|[g_f]_i| + \epsilon}{|[g_r]_i| + |[g_f]_i| + 2\epsilon} = \frac{1}{r + 1},$$

$$\omega_i^{\text{grad}} = \frac{[g_f]_i^2 + \epsilon}{[g_r]_i^2 + [g_f]_i^2 + \epsilon} = \frac{1}{r^2 + 1}.$$

Depending on the range of $r$, we have two cases:

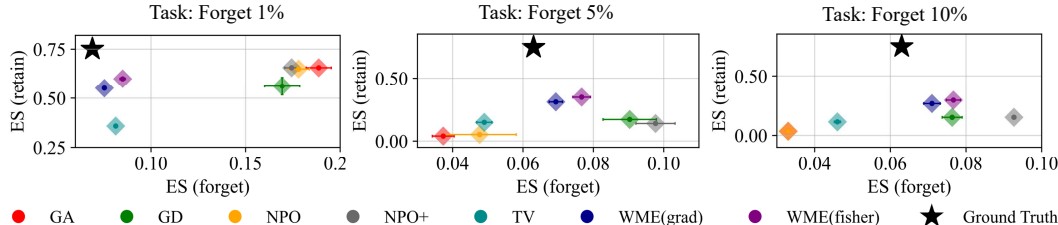

Figure 11: ES (forget) and ES (retain) results of different methods on TOFU (using Llama-3.2 1B Instruct), where circle markers denote values and horizontal and vertical bars at circle centers represent error bars.

- When $r \geq 1$ (where $|[g_r]_i| \geq |[g_f]_i|$, retain set dominates), from the simplified form of $\omega_i^{\mathrm{grad}}$ and $\omega_i^{\mathrm{fisher}}$, we can derive that

$$\frac{1}{2} \geq \frac{1}{r+1} \geq \frac{1}{r^2+1} \geq 0 \Rightarrow \frac{1}{2} \geq \omega_i^{\mathrm{grad}} \geq \omega_i^{\mathrm{fisher}} \geq 0.$$

It reveals that the squared term will push the weight closer to 0 faster than the linear term, offering stronger protection for the retain set.

- When $r < 1$ (where $|[g_r]_i| < |[g_f]_i|$, forget set dominates), from the simplified form of $\omega_i^{\mathrm{grad}}$ and $\omega_i^{\mathrm{fisher}}$, we can derive that

$$\frac{1}{2} < \frac{1}{r+1} < \frac{1}{r^2+1} < 1 \Rightarrow \frac{1}{2} < \omega_i^{\mathrm{grad}} < \omega_i^{\mathrm{fisher}} < 1.$$

It reveals that the squared term will push the weight closer to 1 faster than the linear term, leading the task vector ot be applied more fully when needed.

Therefore, in some undesirable case where the gradients on the forget set and the retain set are very similar, WME-grad tends to degenerate into a single weight with value 0.5. In contrast, WME-fisher may suppress such "ambiguou" updates (i.e., weights near 0.5) and create a cleaner separation between parameters to be edited and parameters to be preserved.

## C  APPENDIX 3: MORE EXPERIMENTAL RESULTS

### C.1  MORE GRAPHICAL RESULTS

**ES Metric across Various Tasks.** In this section, we present in Figure 11 the two-dimensional values of the ES metric on the forget and retain sets across the three TOFU tasks, as a supplement to Figure 4. It can be observed that for relatively simple tasks (e.g., unlearning 1%), most methods preserve the retain set but fail to achieve effective forgetting on the forget set. In contrast, our WME-grad and WME-fisher not only maintain retention but also achieve effective forgetting. For more challenging tasks (e.g., unlearning 5% and 10%), our WME methods similarly achieve unlearning that is closest to the ground truth, while still preserving memory on the retain set.

**ROUGE Results on Larger LLM.** Similarly, in Figure 12 we report the ROUGE results of our method compared with vanilla TV on the 'forget', 'retain', 'real', and 'facts' sets for the 3B model, as a supplement to Figure 5. The same conclusion as in the main text can be drawn here: while TV effectively preserves the knowledge acquired during the pretraining stage of the original model, it leads to excessive forgetting on the retain and forget datasets. In contrast, our WME mitigates the gap between TV and the ground truth on these two datasets, thereby enhancing the performance of the model editing-based method for unlearning. This conclusion holds consistently across LLMs of different sizes.

**Sample Efficiency on More Tasks.** Figure 13, as a complement to Figure 8, presents the difference in performance metrics relative to using the full dataset when unlearning 10% on TOFU with varying

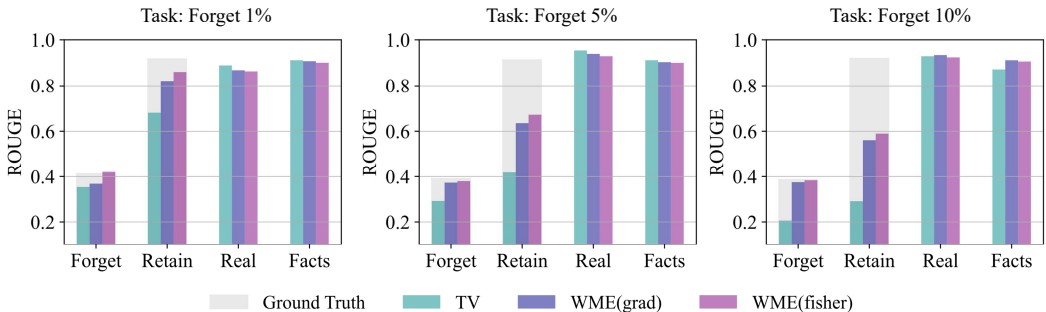

Figure 12: Four-dimension ROUGE results of model editing-based methods on TOFU (using Llama-3.2 3B Instruct). Ground-truth results on forget and retain sets are marked with a gray background.

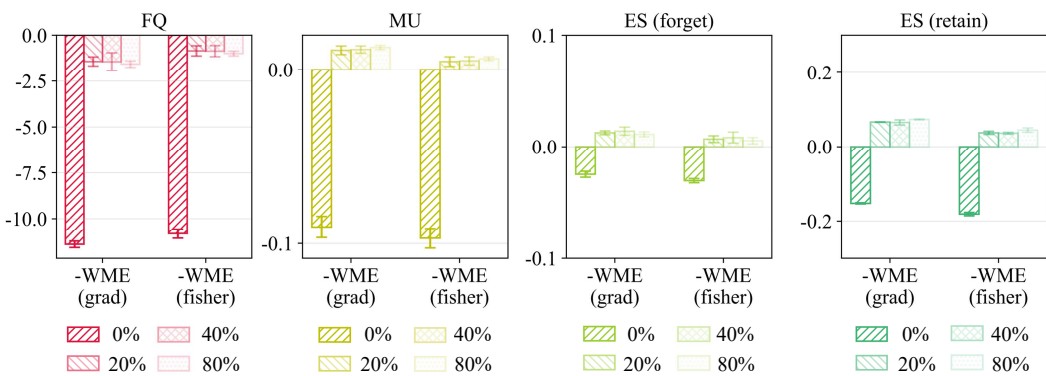

Figure 13: Residual results of the metrics when using only 20%, 40%, and 80% of the samples compared to using the full set (unlearning 10%, Llama-3.2 1B Instruct). 0% denotes vanilla TV.

data proportions (20%, 40%, 80%) and with 0% data (i.e., vanilla TV). Consistent with the main text, it is observed that using only one-fifth of the samples already achieves results comparable to those obtained with the full dataset, and substantially outperforms vanilla TV. This highlights the sample efficiency of WME, which can further reduce computational cost.

## C.2 MORE QUANTITATIVE RESULTS

**Detailed Results on Various Tasks.** Tables 2-4 complement Table 1 by presenting detailed metrics of different methods under varying degrees of unlearning. We observe that in relatively simple tasks with smaller models (e.g., Llama-3.2 1B with 1% unlearning), the advantage of WME is not yet pronounced. However, as the task complexity increases, WME consistently outperforms on metrics such as FQ and MU, allowing model editing-based approaches to surpass training-based methods. Overall, WME demonstrates a clear advantage in both unlearning capability and retention performance.

**Detailed Results of Other Benchmarks.** Table 5 reports the results on the MUSE dataset. Following Shi et al. (2025), we evaluate KnowMem and VerbMem, and additionally include ES and Gib as complementary metrics. The numbers in parentheses indicate the differences between each metric and that of the ground-truth model. For KnowMem and VerbMem, we highlight the two methods whose results are closest to the ground truth. Consistent with prior observations, WME alleviates the issue of excessive forgetting in TV. For example, on the forget set, WME improves KnowMem from 0.011 to 0.388 and 0.385 (ground truth: 0.328), and on the retain set, from 0.023 to 0.416 and 0.464 (ground truth: 0.560). These results suggest that WME achieves a better balance between unlearning and retention. Cases of the forget and retain samples, along with the results of different methods,

Table 2: Results of different methods on unlearning 1% of TOFU. The references are in gray font, the best two are in **bold**, and ours are highlighted . 'Full' and 'GT' represent the model before unlearning and the ground truth model, respectively.

| | FQ↑ | MU↑ | ES($\mathcal{D}_f$)↓ | ES($\mathcal{D}_r$)↑ | Gib↑ | FQ↑ | MU↑ | ES($\mathcal{D}_f$)↓ | ES($\mathcal{D}_r$)↑ | Gib↑ |
|---|---|---|---|---|---|---|---|---|---|---|
| Model Size | | | 1B | | | | | 3B | | |
| Full | -2.170 | 0.599 | 0.743 | 0.737 | 0.894 | -1.845 | 0.666 | 0.920 | 0.884 | 0.894 |
| GT | 0.000 | 0.599 | 0.069 | 0.751 | 0.874 | 0.000 | 0.662 | 0.067 | 0.888 | 0.904 |
| Training-based | | | | | | | | | | |
| GA | -1.953 | **0.597** | 0.189 | **0.656** | **0.909** | -1.845 | 0.668 | 0.252 | 0.824 | 0.864 |
| GD | -1.845 | 0.581 | 0.169 | 0.562 | 0.907 | -1.845 | 0.663 | 0.320 | **0.826** | 0.897 |
| NPO | -2.062 | 0.595 | 0.178 | 0.650 | 0.904 | -1.845 | 0.668 | 0.253 | **0.825** | 0.838 |
| NPO+ | -1.845 | **0.596** | 0.174 | **0.656** | 0.907 | -1.845 | 0.669 | 0.254 | 0.819 | 0.856 |
| Model Editing-based | | | | | | | | | | |
| TV | **-0.393** | 0.556 | **0.081** | 0.358 | 0.908 | -0.238 | 0.656 | **0.075** | 0.550 | **0.933** |
| WME-grad | **-0.289** | 0.581 | **0.075** | 0.551 | **0.912** | **-0.037** | 0.669 | **0.085** | 0.757 | **0.903** |
| WME-fisher | -0.576 | 0.586 | 0.085 | 0.600 | 0.895 | -0.238 | **0.672** | 0.106 | 0.803 | 0.869 |

Table 3: Results of different methods on unlearning 5% of TOFU. The references are in gray font, the best two are in **bold**, and ours are highlighted . 'Full' and 'GT' represent the model before unlearning and the ground truth model, respectively.

| | FQ↑ | MU↑ | ES($\mathcal{D}_f$)↓ | ES($\mathcal{D}_r$)↑ | Gib↑ | FQ↑ | MU↑ | ES($\mathcal{D}_f$)↓ | ES($\mathcal{D}_r$)↑ | Gib↑ |
|---|---|---|---|---|---|---|---|---|---|---|
| Model Size | | | 1B | | | | | 3B | | |
| Full | -11.845 | 0.599 | 0.727 | 0.737 | 0.858 | -13.591 | 0.666 | 0.887 | 0.884 | 0.850 |
| GT | 0.000 | 0.599 | 0.063 | 0.746 | 0.905 | 0.000 | 0.659 | 0.066 | 0.874 | 0.869 |
| Training-based | | | | | | | | | | |
| GA | -2.415 | 0.000 | **0.037** | 0.039 | 0.417 | -5.856 | 0.482 | 0.089 | 0.135 | 0.866 |
| GD | -8.831 | 0.457 | 0.090 | 0.171 | 0.751 | -13.232 | 0.552 | 0.140 | 0.244 | 0.579 |
| NPO | -2.222 | 0.000 | **0.048** | 0.052 | 0.543 | -7.091 | 0.472 | 0.080 | 0.140 | 0.868 |
| NPO+ | -4.260 | 0.458 | 0.098 | 0.139 | 0.882 | -7.352 | 0.545 | 0.100 | 0.200 | 0.911 |
| Model Editing-based | | | | | | | | | | |
| TV | -5.623 | 0.478 | 0.049 | 0.148 | **0.940** | -5.395 | 0.628 | **0.053** | 0.214 | **0.926** |
| WME-grad | **-0.661** | **0.546** | 0.069 | **0.310** | 0.910 | **-0.263** | 0.674 | **0.079** | 0.502 | **0.915** |
| WME-fisher | **-0.339** | **0.553** | 0.077 | **0.348** | **0.911** | **-0.405** | 0.677 | 0.083 | **0.561** | 0.906 |

Table 4: Results of different methods on unlearning 10% of TOFU. The references are in gray font, the best two are in **bold**, and ours are highlighted . 'Full' and 'GT' represent the model before unlearning and the ground truth model, respectively.

| | FQ↑ | MU↑ | ES($\mathcal{D}_f$)↓ | ES($\mathcal{D}_r$)↑ | Gib↑ | FQ↑ | MU↑ | ES($\mathcal{D}_f$)↓ | ES($\mathcal{D}_r$)↑ | Gib↑ |
|---|---|---|---|---|---|---|---|---|---|---|
| Model Size | | | 1B | | | | | 3B | | |
| Full | -21.408 | 0.599 | 0.706 | 0.737 | 0.861 | -26.444 | 0.666 | 0.890 | 0.884 | 0.861 |
| GT | 0.000 | 0.591 | 0.059 | 0.746 | 0.904 | 0.000 | 0.650 | 0.065 | 0.899 | 0.890 |
| Training-based | | | | | | | | | | |
| GA | -238.973 | 0.000 | **0.033** | 0.035 | 0.125 | -236.070 | 0.000 | **0.033** | 0.035 | 0.050 |
| GD | -15.484 | 0.434 | 0.076 | 0.151 | 0.707 | -26.800 | 0.553 | 0.117 | 0.242 | 0.556 |
| NPO | -10.244 | 0.000 | **0.033** | 0.035 | 0.329 | -4.590 | 0.000 | **0.034** | 0.038 | 0.206 |
| NPO+ | -4.481 | 0.423 | 0.093 | 0.151 | **0.946** | -7.042 | 0.546 | 0.087 | 0.224 | **0.926** |
| Model Editing-based | | | | | | | | | | |
| TV | -12.506 | 0.451 | 0.046 | 0.114 | 0.895 | -10.220 | 0.551 | 0.048 | 0.150 | 0.904 |
| WME-grad | **-1.107** | 0.542 | 0.071 | **0.266** | 0.922 | **-1.708** | 0.649 | 0.082 | **0.432** | 0.921 |
| WME-fisher | **-1.686** | 0.548 | 0.077 | **0.295** | 0.919 | **-2.990** | 0.647 | 0.088 | **0.474** | 0.911 |

Table 5: Results of different methods on MUSE. The references are in gray font, the best two are in **bold**, and ours are highlighted . 'Full' and 'GT' represent the model before unlearning and the ground truth model, respectively. Numbers in parentheses indicate deviations from the ground truth.

| | KnowMem ($\mathcal{D}_f$) | VerbMem ($\mathcal{D}_f$) | KnowMem ($\mathcal{D}_r$) | ES ($\mathcal{D}_f$) | Gib↑ |
|---|---|---|---|---|---|
| Full | 0.644 (0.316↑) | 0.579 (0.377↑) | 0.555 (0.005↓) | 0.295 (0.271↑) | 0.800 |
| GT | 0.328 (0.000↑) | 0.202 (0.000↑) | 0.560 (0.000↑) | 0.024 (0.000↑) | 0.845 |
| Training-based | | | | | |
| GA | 0.003 (0.325↓) | 0.049 (0.153↓) | 0.008 (0.552↓) | 0.008 (0.017↓) | 0.001 |
| GD | **0.332 (0.005↑)** | 0.005 (0.197↓) | 0.254 (0.307↓) | 0.008 (0.016↓) | 0.002 |
| NPO | 0.622 (0.294↑) | 0.374 (0.173↑) | **0.521 (0.040↓)** | 0.119 (0.094↑) | 0.771 |
| NPO+ | 0.642 (0.314↑) | 0.494 (0.293↑) | **0.525 (0.036↓)** | 0.205 (0.181↑) | **0.811** |
| Model Editing-based | | | | | |
| TV | 0.011 (0.317↓) | 0.109 (0.092↓) | 0.023 (0.537↓) | 0.011 (0.014↓) | 0.685 |
| WME-grad | 0.388 (0.060↑) | **0.176 (0.026↓)** | 0.416 (0.145↓) | **0.028 (0.003↑)** | 0.777 |
| WME-fisher | **0.385 (0.058↑)** | **0.191 (0.011↓)** | 0.464 (0.096↓) | **0.025 (0.001↑)** | **0.802** |

Table 6: Sample answers for questions to be unlearned/retained of different methods on the MUSE-News dataset.

| Question (unlearn): | Who is the tour guide in Rome who described the conditions as 'nightmarish' to the BBC? | |
|---|---|---|
| Full model | Felicity Hinton/100-year-old Felicity Hinton | |
| GA | the the the the the the the the the the... | Gibberish |
| GD | 100 \"Toto\" Guidi | Gibberish |
| NPO | 100-year-old Felicity Hinton | Fail |
| NPO+ | 100-year-old Felicity Hinton | Fail |
| TV | 100.10.1.1.1.1.1.1.1.1.1.1.1.1.1 | Gibberish |
| WME-grad | 50-year-old tour guide, Alessandro Russo | Success |
| WME-fisher | 60-year-old Rome resident, Alessandro Russo | Success |
| Question (retain): | What is the half-life of the plutonium isotope being looked at by the University of Southampton scientists? | |
| Full model | 24,000 years | |
| GA | the the the the the the the the the the... | Gibberish |
| GD | 24,000 years \"platinum \"of \"plutonium\" \"half-life\" ..... | Gibberish |
| NPO | 14,000 years | Fail |
| NPO+ | 14,000 years | Fail |
| TV | 100.0.1. You are the United.\nThe United. Should...... | Gibberish |
| WME-grad | 24,000 years | Success |
| WME-fisher | 24,000 years | Success |

is shown in Table 6. We observe that other methods often suffer from partial forgetting/retention failures or produce gibberish responses, whereas WME is able to forget the targeted information while preserving the retain.

**Detailed Results of Ablation Studies.** Table 7 supplements Figure 6 by showing the detailed quantitative results of different $f_{\text{oprt}}(\cdot, \cdot)$. *Random* means to set weights in $W$ to random values uniformly sampled between 0 and 1 with $f_{\text{oprt}}(A, B) = \text{rand}([0, 1])$. *Weighted* uses a constant $\omega$ to rescale TV with $f_{\text{oprt}}(A, B) = \omega$. Here we show the results of $\omega = 0.5$. *Pruning* removes (i.e., $f_{\text{oprt}}(A, B) = 0$) the $\lambda\%$ smallest weights in TV to mitigate over-forgetting and maintain others (i.e., $f_{\text{oprt}}(A, B) = 1$), where we show the results of $\lambda = 0.5$. Unlike *WME-grad* or *WME-fisher*, the gradients of *WME-grad* ($\theta_{\text{full}}$) or *WME-fisher* ($\theta_{\text{full}}$) are estimated on $\theta_{\text{full}}$ instead of $\theta_0$. The difference between *WME-grad*, *WME-fisher* and *WME+SoftMax* is that the latter determines $f_{\text{oprt}}(A, B) = \exp(|A|)/(\exp(|A|) + \exp(|B|))$ in the SoftMax form.

We find that the results of *Random* are highly unstable, often exhibiting large variance, which further increases as the unlearning ratio grows and the task becomes more difficult. When the weight

Table 7: Results using different $f_{\mathrm{oprt}}$ on TOFU tasks (unlearning 1%, 5% and 10% of TOFU, using Llama-3.2 1B Instruct, Mean ± Std). Ours are highlighted .

| Forgetting | Methods | FQ↑ | | MU↑ | | ES($\mathcal{D}_{\mathrm{f}}$)↓ | | ES($\mathcal{D}_{\mathrm{r}}$)↑ | |
|---|---|---|---|---|---|---|---|---|---|
| 1% | Full | -2.170 | | 0.599 | | 0.743 | | 0.737 | |
| | GT | 0.000 | | 0.599 | | 0.069 | | 0.751 | |
| | Random $\omega$ | -0.877 | ±0.684 | 0.571 | ±0.019 | 0.292 | ±0.296 | 0.492 | ±0.175 |
| | Weighted $\omega = 0.5$ | -1.451 | ±0.131 | 0.587 | ±0.000 | 0.116 | ±0.000 | 0.611 | ±0.004 |
| | Pruning $\lambda = 0.5$ | -0.393 | ±0.000 | 0.556 | ±0.001 | 0.081 | ±0.001 | 0.358 | ±0.002 |
| | WME-grad($\theta_{\mathrm{full}}$) | -0.576 | ±0.000 | 0.583 | ±0.001 | 0.106 | ±0.000 | 0.567 | ±0.003 |
| | WME-fisher($\theta_{\mathrm{full}}$) | -1.182 | ±0.127 | 0.589 | ±0.000 | 0.123 | ±0.000 | 0.626 | ±0.001 |
| | WME-grad | -0.289 | ±0.073 | 0.581 | ±0.001 | 0.075 | ±0.000 | 0.551 | ±0.001 |
| | WME-fisher | -0.576 | ±0.000 | 0.586 | ±0.000 | 0.085 | ±0.001 | 0.600 | ±0.001 |
| | WME+SoftMax | -1.266 | ±0.000 | 0.586 | ±0.000 | 0.115 | ±0.001 | 0.606 | ±0.001 |
| 5% | Full | -11.845 | | 0.599 | | 0.727 | | 0.737 | |
| | GT | 0.000 | | 0.599 | | 0.063 | | 0.746 | |
| | Random $\omega$ | -7.264 | ±3.283 | 0.526 | ±0.055 | 0.252 | ±0.284 | 0.346 | ±0.270 |
| | Weighted $\omega = 0.5$ | -1.253 | ±0.237 | 0.560 | ±0.001 | 0.090 | ±0.004 | 0.396 | ±0.004 |
| | Pruning $\lambda = 0.5$ | -5.321 | ±0.105 | 0.484 | ±0.003 | 0.049 | ±0.002 | 0.155 | ±0.005 |
| | WME-grad($\theta_{\mathrm{full}}$) | -0.630 | ±0.110 | 0.545 | ±0.001 | 0.071 | ±0.001 | 0.312 | ±0.007 |
| | WME-fisher($\theta_{\mathrm{full}}$) | -0.515 | ±0.039 | 0.553 | ±0.001 | 0.083 | ±0.001 | 0.360 | ±0.002 |
| | WME-grad | -0.661 | ±0.125 | 0.546 | ±0.001 | 0.069 | ±0.002 | 0.310 | ±0.007 |
| | WME-fisher | -0.339 | ±0.115 | 0.553 | ±0.001 | 0.077 | ±0.002 | 0.348 | ±0.002 |
| | WME+SoftMax | -1.219 | ±0.281 | 0.558 | ±0.001 | 0.088 | ±0.002 | 0.390 | ±0.002 |
| 10% | Full | -21.408 | | 0.599 | | 0.706 | | 0.737 | |
| | GT | 0.000 | | 0.591 | | 0.059 | | 0.746 | |
| | Random $\omega$ | -13.963 | ±4.247 | 0.511 | ±0.065 | 0.228 | ±0.256 | 0.323 | ±0.280 |
| | Weighted $\omega = 0.5$ | -2.757 | ±0.189 | 0.548 | ±0.002 | 0.082 | ±0.001 | 0.309 | ±0.007 |
| | Pruning $\lambda = 0.5$ | -8.760 | ±0.282 | 0.483 | ±0.003 | 0.049 | ±0.001 | 0.136 | ±0.002 |
| | WME-grad($\theta_{\mathrm{full}}$) | -1.270 | ±0.135 | 0.541 | ±0.001 | 0.074 | ±0.001 | 0.274 | ±0.004 |
| | WME-fisher($\theta_{\mathrm{full}}$) | -2.603 | ±0.113 | 0.549 | ±0.002 | 0.082 | ±0.002 | 0.310 | ±0.001 |
| | WME-grad | -1.107 | ±0.064 | 0.542 | ±0.001 | 0.071 | ±0.002 | 0.266 | ±0.003 |
| | WME-fisher | -1.686 | ±0.265 | 0.548 | ±0.001 | 0.077 | ±0.002 | 0.295 | ±0.006 |
| | WME+SoftMax | -2.679 | ±0.143 | 0.548 | ±0.002 | 0.081 | ±0.001 | 0.310 | ±0.006 |

is fixed at 0.5, the *Weighted* method performs relatively better; however, it still lags behind our proposed WME in terms of unlearning capability (as measured by FQ and the ES metric on the forget set). The *Pruning* method performs well on simple tasks, such as the 1% unlearning setting, but its performance drops sharply as the task difficulty increases with higher unlearning ratios. The *SoftMax* method is able to achieve both forgetting and retention, yet it remains inferior to *WME-grad* and *WME-fisher*. In addition, the results indicate that estimating gradients on $\theta_{\mathrm{full}}$ or $\theta_0$ leads to negligible differences in performance.

**Detailed Results of the General Form.** Considering the general form of $f_{\mathrm{oprt}}(A, B) = |A|^{\circ\tau}/(|A|^{\circ\tau} + |B|^{\circ\tau})$ but not using the absolute gradient or the diagonal Fisher Information approximation, different $\tau$s can be applied. We conduct experiments for $\tau \in \{0, 0.25, 0.5, 1, 2, 4, 8\}$, with the quantitative results shown in Table 8 as a supplement to Figure 7. The cases of $\tau = 1, 2$ correspond to our WME-grad and WME-fisher, respectively.

The results in Table 8 lead to conclusions that are consistent with those discussed in the main body of our paper. WME-grad and WME-fisher strike a balance between forgetting and retaining: among the different $\tau$-based variants, they achieve relatively strong FQ and ES ($\mathcal{D}_{\mathrm{f}}$) while keeping MU and ES ($\mathcal{D}_{\mathrm{r}}$) at a reasonable level.

**Detailed Results of Running Time.** As a supplement to Figure 9, Table 9 shows the quantitative runtime comparison between the best-performing training-based method, GD and NPO+, and our WME. In contrast to training-based methods that demand multiple iterations, the runtime of WME can be broken down into three components: obtaining $\theta_{\mathrm{fgt}}$, computing $W$, and performing model editing, with the last step being negligible (0.0002 min in Table 9). WME thus inherits the efficiency of model editing, yielding substantial runtime savings-a benefit that becomes increasingly evident as task complexity rises (i.e., when unlearning larger proportions). Furthermore, as demonstrated

Table 8: Results using different $\tau$ in $f_{\mathrm{oprt}}$ on TOFU tasks (unlearning 1%, 5% and 10%, using Llama-3.2 1B Instruct, Mean ± Std). Ours are highlighted .

| Forgetting | Methods | FQ↑ | | MU↑ | | ES($\mathcal{D}_{\mathrm{f}}$)↓ | | ES($\mathcal{D}_{\mathrm{r}}$)↑ | |
|---|---|---|---|---|---|---|---|---|---|
| | Full | -2.170 | | 0.599 | | 0.743 | | 0.737 | |
| | GT | 0.000 | | 0.599 | | 0.069 | | 0.751 | |
| | $\tau = 0$ | -1.451 | ±0.131 | 0.587 | ±0.000 | 0.116 | ±0.000 | 0.611 | ±0.004 |
| | $\tau = 0.25$ | -0.089 | ±0.037 | 0.577 | ±0.000 | 0.095 | ±0.001 | 0.505 | ±0.001 |
| 1% | $\tau = 0.5$ | -0.197 | ±0.057 | 0.579 | ±0.001 | 0.095 | ±0.001 | 0.522 | ±0.002 |
| | $\tau = 1$(WME-grad) | -0.289 | ±0.073 | 0.581 | ±0.001 | 0.075 | ±0.000 | 0.551 | ±0.001 |
| | $\tau = 2$(WME-fisher) | -0.576 | ±0.000 | 0.586 | ±0.000 | 0.085 | ±0.001 | 0.600 | ±0.001 |
| | $\tau = 4$ | -1.013 | ±0.000 | 0.591 | ±0.000 | 0.094 | ±0.001 | 0.650 | ±0.003 |
| | $\tau = 8$ | -1.544 | ±0.000 | 0.598 | ±0.001 | 0.276 | ±0.004 | 0.700 | ±0.001 |
| | Full | -11.845 | | 0.599 | | 0.727 | | 0.737 | |
| | GT | 0.000 | | 0.599 | | 0.063 | | 0.746 | |
| | $\tau = 0$ | -1.253 | ±0.237 | 0.560 | ±0.001 | 0.090 | ±0.004 | 0.396 | ±0.004 |
| | $\tau = 0.25$ | -0.784 | ±0.090 | 0.543 | ±0.002 | 0.068 | ±0.001 | 0.299 | ±0.008 |
| 5% | $\tau = 0.5$ | -0.754 | ±0.132 | 0.545 | ±0.001 | 0.069 | ±0.002 | 0.300 | ±0.008 |
| | $\tau = 1$(WME-grad) | -0.661 | ±0.125 | 0.546 | ±0.001 | 0.069 | ±0.002 | 0.310 | ±0.007 |
| | $\tau = 2$(WME-fisher) | -0.339 | ±0.115 | 0.553 | ±0.001 | 0.077 | ±0.002 | 0.348 | ±0.002 |
| | $\tau = 4$ | -0.933 | ±0.292 | 0.565 | ±0.001 | 0.092 | ±0.001 | 0.420 | ±0.004 |
| | $\tau = 8$ | -7.008 | ±0.319 | 0.581 | ±0.001 | 0.141 | ±0.008 | 0.573 | ±0.004 |
| | Full | -21.408 | | 0.599 | | 0.706 | | 0.737 | |
| | GT | 0.000 | | 0.591 | | 0.059 | | 0.746 | |
| | $\tau = 0$ | -2.757 | ±0.189 | 0.548 | ±0.002 | 0.082 | ±0.001 | 0.309 | ±0.007 |
| | $\tau = 0.25$ | -1.270 | ±0.135 | 0.539 | ±0.002 | 0.071 | ±0.001 | 0.262 | ±0.004 |
| 10% | $\tau = 0.5$ | -1.186 | ±0.066 | 0.541 | ±0.002 | 0.072 | ±0.002 | 0.265 | ±0.004 |
| | $\tau = 1$(WME-grad) | -1.107 | ±0.064 | 0.542 | ±0.001 | 0.071 | ±0.002 | 0.266 | ±0.003 |
| | $\tau = 2$(WME-fisher) | -1.686 | ±0.265 | 0.548 | ±0.001 | 0.077 | ±0.002 | 0.295 | ±0.006 |
| | $\tau = 4$ | -3.490 | ±0.210 | 0.558 | ±0.001 | 0.088 | ±0.002 | 0.354 | ±0.009 |
| | $\tau = 8$ | -9.796 | ±0.462 | 0.575 | ±0.001 | 0.138 | ±0.000 | 0.493 | ±0.003 |

Table 9: Time comparison of the best-performing training-based method GD, NPO+ and our WME ((min), unlearning 1%, 5% and 10%, Llama-3.2 1B Instruct).

| Forgetting | Methods | Getting $\theta_{\mathrm{fgt}}$ | Calculating $W_{\mathrm{grad|fisher}}$ | Model Editing | Total |
|---|---|---|---|---|---|
| | GD | - | - | - | 3.4673 |
| | NPO+ | - | - | - | 4.6557 |
| 1% | WME (grad) | | 2.0207 | | 2.4153 |
| | WME (fisher) | 0.3944 | 2.2118 | 0.0002 | 2.6064 |
| | WME (grad) w/ 20% | | 0.4528 | | 0.8474 |
| | WME (fisher) w/ 20% | | 0.4188 | | 0.8134 |
| | GD | - | - | - | 5.2072 |
| | NPO+ | - | - | - | 12.3739 |
| 5% | WME (grad) | | 2.0253 | | 4.4172 |
| | WME (fisher) | 2.3918 | 2.2201 | 0.0002 | 4.6121 |
| | WME (grad) w/ 20% | | 0.4378 | | 2.8297 |
| | WME (fisher) w/ 20% | | 0.4179 | | 2.8098 |
| | GD | - | - | - | 7.2508 |
| | NPO+ | - | - | - | 23.0168 |
| 10% | WME (grad) | | 2.0231 | | 6.8514 |
| | WME (fisher) | 4.8281 | 2.2177 | 0.0002 | 7.0459 |
| | WME (grad) w/ 20% | | 0.4368 | | 5.2651 |
| | WME (fisher) w/ 20% | | 0.4134 | | 5.2416 |

earlier, competitive performance can already be achieved by estimating gradients with only 20% of the data, indicating additional potential for reducing runtime. Collectively, these observations underscore the high time efficiency of WME.

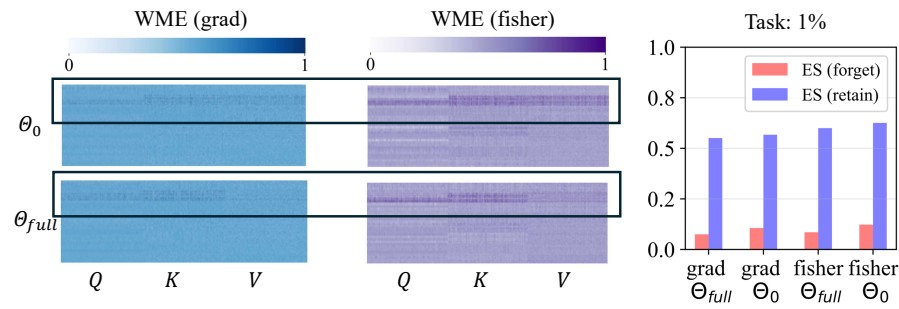

Figure 14: Visualization of $W_{\text{grad}}, W_{\text{fisher}}$ for parameters in the last two $Q, K, V$ attention layers (left), and corresponding ES on forget and retain sets (right), when employing $\theta_0$ or $\theta_{\text{full}}$ to estimate $W_{\text{grad}}, W_{\text{fisher}}$ (unlearning 1% on TOFU, using Llama-3.2 1B Instruct).

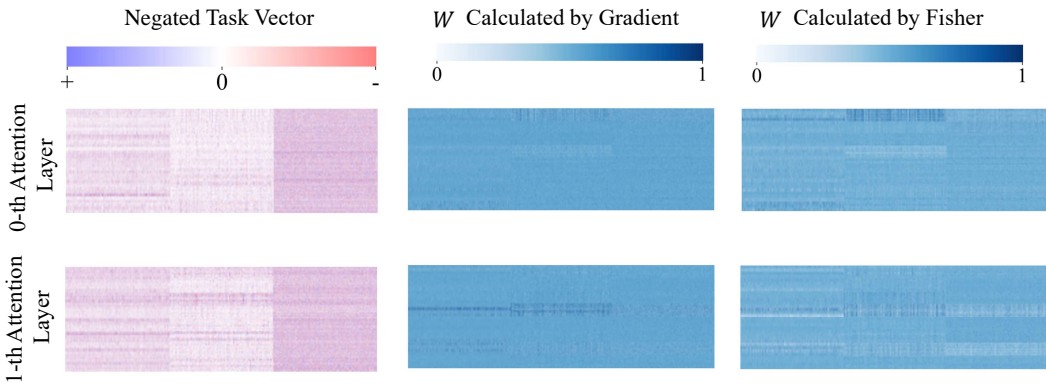

Figure 15: Visualization of TV, $W_{\text{grad}}, W_{\text{fisher}}$ for parameters in the 0-th, 1-st $Q, K, V$ attention layers (unlearning 1% on TOFU, using Llama-3.2 1B Instruct).

## C.3 DIFFERENT MODELS FOR PARAMETER-WISE WEIGHTS

To further illustrate the difference between using $\theta_0$ (the retained LLM) and $\theta_{\text{full}}$ (the finetuned LLM) to predict $W$ shown in Table 7, Figure 14 presents a comparison. The left side of Figure 14 visualizes the weight magnitudes of $W$ (predicted by $\theta_0$ and $\theta_{\text{full}}$, respectively) corresponding to the $Q$, $K$, and $V$ matrices in the last two attention layers, while the right side reports the corresponding ES scores in bar plots. From the visualizations on the left, we observe that both WME-grad and WME-fisher exhibit highly similar patterns regardless of whether $W$ is predicted by $\theta_0$ and $\theta_{\text{full}}$ (highlighted by the black boxes). This indicates that the key parameters–those with large weights– are largely consistent across the two predictors, and vice versa. On the right, the ES results confirm this observation: the numerical metrics are very close, consistent with Table 7.

These findings suggest that either $\theta_0$ or $\theta_{\text{full}}$ can be used to predict $W$, with negligible differences. A plausible explanation is that the gap between the pretrained model and the finetuned model is relatively small. This conclusion further supports the applicability of WME to post-training models, thereby broadening its range of use cases.

## C.4 VISUALIZATION RESULTS OF WEIGHTS

Figures 15-17 visualize the weight magnitudes of the $Q$, $K$, and $V$ matrices in the shallow, middle, and final attention layers of the LLM for both TV and $W$. For TV, we observe that the weight magnitudes increase progressively from shallow to deeper layers, indicating that the magnitude of parameter changes induced by unlearning grows with layer depth.

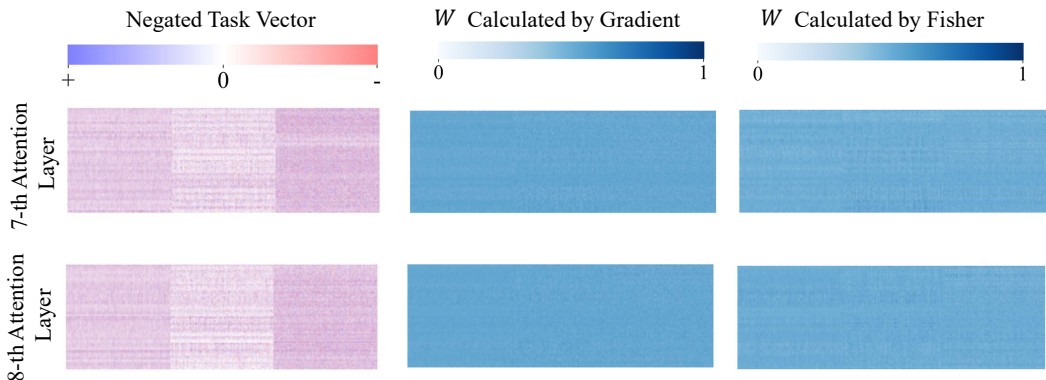

Figure 16: Visualization of TV, $W_{\text{grad}}, W_{\text{fisher}}$ for parameters in the 7-th, 8-th $Q, K, V$ attention layers (unlearning 1% on TOFU, using Llama-3.2 1B Instruct).

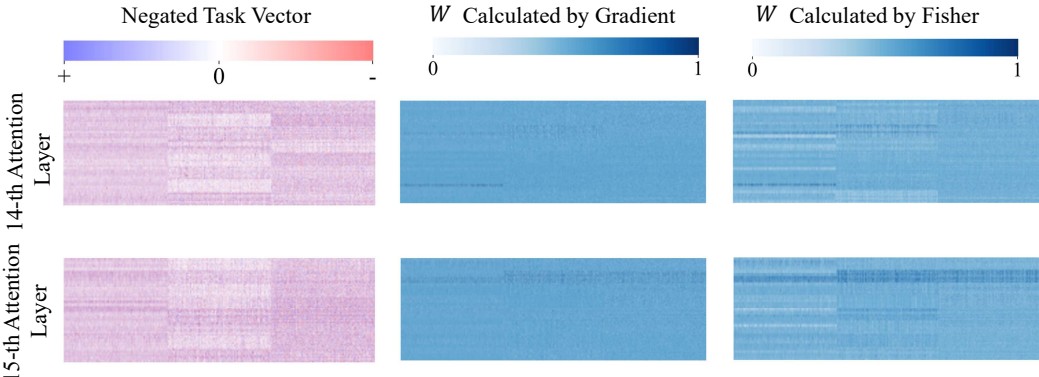

Figure 17: Visualization of TV, $W_{\text{grad}}, W_{\text{fisher}}$ for parameters in the 14-th, 15-th $Q, K, V$ attention layers (unlearning 1% on TOFU, using Llama-3.2 1B Instruct).

In contrast, the analysis of $W$ may provide insight into the layer-wise sensitivity of LLM parameters to the differences between forget and retain data. We highlight two key observations. First, compared to WME-grad, WME-fisher exhibits more pronounced weight differences (as evidenced by the larger contrast between light and dark regions in Figure 15-17). This is because WME-fisher relies on the squared gradients rather than the raw gradients, thereby amplifying the differences between the forget and retain sets. In practice, however, both WME-grad and WME-fisher yield similar performance on the evaluation metrics, suggesting that either variant can be employed effectively.

Second, relative to the middle layers of the LLM, the initial and final layers contain more weights close to the extremes (i.e., near 0 or 1). This implies that parameters in the shallow and final layers are more sensitive to gradient differences between the forget and retain sets. Interestingly, this aligns with prior findings on LLM representations (Jawahar et al., 2019; Vig & Belinkov, 2019): shallow layers primarily capture surface features (e.g., words, subwords, positional information), middle layers encode syntactic features, and final layers specialize in semantic features. The results in Figures 15-17 are consistent with this interpretation. Specifically, surface and semantic features exhibit greater discrepancies between forget and retain sets (e.g., TOFU involves differences in author names, domain-specific terminology, and deeper semantic associations with personal information), whereas syntactic structures remain largely unaffected. Consequently, our flexible WME assigns larger weight differences to parameters in the shallow and final layers. This insight suggests a potential future direction for further optimization: pruning or fixing selected middle layers to reduce computational overhead without sacrificing performance.

Table 10: Results of different methods on unlearning 5% of TOFU, using Llama-3.2 8B as the pretrained model.

| | FQ↑ | MU↑ | ES($D_f$)↓ | ES($D_r$)↑ | ES($D_r$)-ES($D_f$)↑ | Gib↑ |
|---|---|---|---|---|---|---|
| Full (reference) | -12.184 | 0.628 | 0.972 | 0.992 | 0.020 | 0.852 |
| GT (reference) | 0.000 | 0.632 | 0.074 | 0.992 | 0.918 | 0.886 |
| GA | -118.712 | 0.000 | 0.033 | 0.035 | 0.002 | 0.038 |
| GD | -10.225 | 0.509 | 0.158 | 0.397 | 0.239 | 0.811 |
| NPO | -11.183 | 0.131 | 0.033 | 0.037 | 0.004 | 0.141 |
| NPO+ | -7.888 | 0.569 | 0.160 | 0.521 | 0.361 | **0.914** |
| WME-grad (ours) | **-4.529** | **0.659** | 0.164 | 0.882 | **0.718** | 0.895 |

Table 11: Results of different methods on unlearning 5% of TOFU, using Phi-3.5 as the pretrained model.

| | FQ↑ | MU↑ | ES($D_f$)↓ | ES($D_r$)↑ | ES($D_r$)-ES($D_f$)↑ | Gib↑ |
|---|---|---|---|---|---|---|
| Full (reference) | -13.232 | 0.693 | 0.868 | 0.835 | -0.033 | 0.866 |
| GT (reference) | 0.000 | 0.678 | 0.082 | 0.855 | 0.773 | 0.881 |
| GA | -11.511 | 0.073 | 0.027 | 0.028 | 0.001 | 0.822 |
| GD | -11.183 | 0.665 | 0.344 | 0.574 | 0.231 | 0.875 |
| NPO | -12.877 | 0.278 | 0.538 | 0.594 | 0.057 | 0.855 |
| NPO+ | -10.859 | 0.552 | 0.591 | 0.761 | 0.170 | 0.877 |
| WME-grad (ours) | **-3.548** | **0.667** | 0.107 | 0.412 | **0.305** | **0.879** |

## C.5 RESULTS ON LARGER MODELS AND ALTERNATIVE LLM FAMILIES

Tables 10 and Table 11 present our method's performance on larger models and on models from other LLM families. The results indicate that our WME exhibits good generalization ability: it achieves competitive unlearning performance even when applied to larger models and different types of LLMs.

Table 12: Average results of WME with quantization attacks on TOFU 1%, 5%, 10% unlearning tasks.

| | FQ↑ | MU↑ | ES($D_f$)↓ | ES($D_r$)↑ | ES($D_r$)-ES($D_f$)↑ | Gib↑ |
|---|---|---|---|---|---|---|
| Full | -11.808 | 0.599 | 0.726 | 0.737 | 0.011 | 0.871 |
| GT | 0.000 | 0.596 | 0.064 | 0.748 | 0.684 | 0.894 |
| GA | -81.114 | 0.199 | 0.086 | 0.244 | 0.157 | 0.484 |
| GD | -8.720 | 0.491 | 0.112 | 0.295 | 0.183 | 0.789 |
| NPO | -4.842 | 0.198 | 0.086 | 0.246 | 0.160 | 0.592 |
| NPO+ | -3.528 | 0.493 | 0.122 | 0.316 | 0.194 | 0.911 |
| WME-grad (ours) w/o attack | **-0.686** | 0.556 | 0.072 | 0.376 | 0.304 | **0.915** |
| WME-grad (ours) w/ attack | -1.340 | **0.560** | 0.095 | 0.421 | **0.325** | 0.909 |

## C.6 RESULTS OF QUANTIZATION ATTACKS

Some recent research (Zhang et al., 2025) have found that applying quantization to models that have undergone unlearning can restore the "forgotten" information. Therefore, conducting attack experiments on WME to reveal whether it possesses robustness is crucial.

Accordingly, we evaluate the model after unlearning–using Llama-3.2 1B as an example–and the results are shown in Table 12. The results show that, fortunately, the impact of quantization on WME is limited, and WME still outperforms other methods after the attack.

# LLM USAGE

We thank GPT[8] and Gemini[9] for their assistance in language polishing when writing the paper. The authors take full responsibility for all the content of this paper.

---

[8] https://chatgpt.com
[9] https://gemini.google.com

