# OpenReview forum: "Parameter-wise Weighted Model Editing for Efficient and Retentive LLM Unlearning"
_ICLR.cc/2026/Conference — Submitted to ICLR 2026_

### Official Review · Reviewer_Qqq2 · 2025-10-29

**Soundness:** 3
**Presentation:** 2
**Contribution:** 3
**Rating:** 6
**Confidence:** 4

**Summary:**

This paper introduces Weighted Model Editing (WME), a method for large language model unlearning. By applying parameter-wise weights to the task vector, WME selectively forgets target information while preserving general knowledge. It outperforms baseline methods and even rivals training-based approaches in effectiveness and utility.

**Strengths:**

1. The problem of LLMs is clear
2. The authors provide some experiments, validating the effectiveness of their method.

**Weaknesses:**

1. The weighting function (corresponding to Equations 4 and 5) currently appears to be more of a heuristic design. The authors do not sufficiently explain why the absolute value of the gradient or the squared ratio is used as the weighting proportion, nor do they provide corresponding theoretical analysis or approximate derivation to support this form.

 2. The authors choose to compute the gradients on the original model parameters theta_0 (rather than the full model theta_full) to estimate the weights W. This decision is somewhat counterintuitive, as the final editing operation (Equation 3) is actually performed on theta_full. The main text does not explain this, and the related discussion (Appendix C.3) is placed in the appendix, which may easily confuse readers.

3. Although TOFU and MUSE are commonly used evaluation benchmarks, they are primarily based on semi-synthetic or fictional data. If the paper could supplement with a small-scale, real-world privacy-related forgetting experiment, even as a simple validation, it would significantly enhance its persuasiveness.

4. As shown in Figure 7 and Algorithm 1, the parameters tau and alpha may have a considerable impact on the results. It is recommended that the authors add a brief explanation in the appendix or main text regarding the selection or tuning of these hyperparameters.

**Questions:**

1. Regarding the weighting function Wi: It is recommended to enhance the methodology section by incorporating an intuitive explanation or theoretical derivation for the design of the weighting function Wi. This would significantly improve the understandability and justification of the proposed approach.

2. Regarding the parameter choice (theta_0 vs. theta_full): The analysis and rationale behind the choice of computing gradients on the original parameters theta_0 instead of the full model theta_full, which is currently in Appendix C.3, should be concisely summarized and integrated into the main body of the paper. This clarification is crucial for reader comprehension, as the choice may appear counterintuitive at first glance.

3. Regarding experimental validation: To further strengthen the practical relevance of the method, consider adding a case study involving real-world or privacy-centric unlearning scenarios. Such an experiment, even on a small scale, would provide valuable validation of the method's applicability beyond synthetic benchmarks.

4. Regarding hyperparameters: The paper would benefit from a brief discussion or guidance on the selection and tuning of the hyperparameters tau and alpha, whose values appear to significantly influence the results as indicated in Figure 7 and Algorithm 1.

---

> ### Author Response · Authors · 2025-11-23
> **Response to Reviewer Qqq2 (Part I)**
>
> **Rebuttal to W1/Q1:** We thank the reviewer for the helpful suggestion.
> The design of our weighting functions in Eqns.(4-5) is not arbitrary, but arises from a straightforward way to measure the *relative importance* of each parameter for forgetting vs. retention by their ratio.
>
> Let $g_f = \nabla \mathcal{L} (D_f; \theta_0)$ and $g_r = \nabla \mathcal{L} (D_r; \theta_0)$ be gradients at parameter $q_i$. Using the magnitudes $|[g_f]_i|$ and $|[g_r]_i|$ as local importance scores, then
> - Eqn.(4) defines $\omega_i^{\rm grad}$ which directly encodes the fraction of gradient mass coming from the forget set.
> - Eqn.(5) replaces gradients by squared gradients; as we have shown in Appendix  B.1, this is exactly the diagonal Fisher information approximation, a standard measure of parameter sensitivity. This gives $\omega_i^{\rm fisher}$ that amplifies differences between $g_f$ and $g_r$.
>
> In Appendix B.2-B.3 we proved that both forms satisfy our desiderata: if $|[g_f]_i| \gg [g_r]_i$, then $\omega_i \to 1$ (task vector is kept); otherwise $\omega_i \to 0$ (task vector is suppressed). Moreover, $\omega_i^{\rm fisher}$ moves away from 0.5 faster as $\frac{|[g_r]_i|}{|[g_f]_i|}$ deviates from 1, leading to more polarized weights and empirically slightly better FQ/MU.
>
> We have an intuitive explanation in Section 4.2. We hope this additional information provides clearer, intuitive and analytical support for the design of our weighting function.
>
> **Rebuttal to W2/Q2:** We appreciate the opportunity to clarify this design choice, and will add the following clarification to the main text to address potential reader concerns: "We choose to estimate gradients on $\theta_0$ because $\theta_0$ is a cleaner model that does not contain training data from forget set or retain set. However, in practice, estimating gradients on $\theta_0$ or on $\theta_{\rm full}$ makes a negligible difference (see Appendix C.3 for a detailed discussion)."

---

> ### Author Response · Authors · 2025-11-23
> **Response to Reviewer Qqq2 (Part II)**
>
> **Rebuttal to W3/Q3:** We sincerely thank the reviewer for the suggestion. We also agree that demonstrating performance on realistic text is valuable. While TOFU is fictitious, the MUSE‑News benchmark we use is built from real BBC news articles published after August 2023. We have prepared a small case study of the forget and retain samples from MUSE‑News, along with the outputs from unlearning methods, in the following table.
>
> | Question (to be unlearned): | Who is the tour guide in Rome who described the conditions as 'nightmarish' to the BBC?                       |           |
> |-------------------------------|---------------------------------------------------------------------------------------------------------------|-----------|
> | Full model                 | Felicity Hinton/100-year-old Felicity Hinton                                                                  |           |
> | GA                            | the the the the the the the the the the…                                                                      | Gibberish |
> | GD                            | 100 \"Toto\" Guidi                                                                                            | Gibberish |
> | NPO                           | 100-year-old Felicity Hinton                                                                                  | Fail      |
> | NPO+                          | 100-year-old Felicity Hinton                                                                                  | Fail      |
> | TV                            | 100.10.1.1.1.1.1.1.1.1.1.1.1.1.1                                                                              | Gibberish |
> | WME-grad                      | 50-year-old tour guide, Alessandro Russo                                                                      | Success   |
> | WME-fisher                    | 60-year-old Rome resident, Alessandro Russo                                                                   | Success   |
>
>
> | Question (to be retained):    | What is the half-life of the plutonium isotope being looked at by the University of Southampton scientists? |           |
> |-------------------------------|---------------------------------------------------------------------------------------------------------------|-----------|
> | Full model                 | 24,000 years                                                                                                  |           |
> | GA                            | the the the the the the the the the the…...                                                                   | Gibberish |
> | GD                            | 24,000 years \"platinum \"of \"plutonium\"   \"half-life\" \"the \"half-life of \"                            | Gibberish |
> | NPO                           | 14,000 years                                                                                                  | Fail      |
> | NPO+                          | 14,000 years                                                                                                  | Fail      |
> | TV                            | 100.0.1. You are the United.\nThe United. Should the United.\nThe United                                      | Gibberish |
> | WME-grad                      | 24,000 years                                                                                                  | Success   |
> | WME-fisher                    | 24,000 years                                                                                                  | Success   |
>
>
> From these results, we observe that other methods often suffer from partial forgetting/retention failures or produce gibberish responses, whereas WME is able to forget the targeted information while preserving the retain set.
> We will add these new results and discussion to Appendix C.2.

---

> ### Author Response · Authors · 2025-11-23
> **Response to Reviewer Qqq2 (Part III)**
>
> **Rebuttal to W4/Q4:** We sincerely thank the reviewer for the question. We will include the hyperparameter details in Appendix A.2. In fact, $\tau=1$ and $\tau=2$ correspond to the two forms in Eqn.(4) and Eqn.(5), respectively; they are fixed choices **rather than tunable** hyperparameters; other $\tau$ values only serve for the ablation (Figure 7), where we discuss alternative forms of weighting functions, showing that these two choices already strike a good balance between FQ, MU, ${\rm ES}(D_f)$, and ${\rm ES}(D_r)$.
>
> The learning rate $\alpha$ is simply the standard fine-tuning learning rate used to obtain $\theta_{\rm fgt}$. For fair comparison, we follow [1] to set the learning rate for *all* methods *without* extra tuning.
> We appreciate the reviewer’s careful reading, and we will revise Appendix A.2 to clarify that WME does not introduce additional tuned hyperparameters beyond those already used for finetuning.
>
>
> [1] Tofu: A task of fictitious unlearning for LLMs.
>
> ---
> We sincerely hope these clarifications address the reviewer’s concerns. If there are any remaining questions, we would be happy to address them. Otherwise, we would appreciate it if the reviewer could reconsider and raise the scores.

---

### Official Review · Reviewer_LDUZ · 2025-10-29

**Soundness:** 2
**Presentation:** 3
**Contribution:** 2
**Rating:** 4
**Confidence:** 3

**Summary:**

This paper presents Parameter-wise Weighted Model Editing (WME), a lightweight approach to selective knowledge unlearning for large language models. Instead of applying a uniform scalar to the task vector, the authors introduce parameter-wise weights computed from the relative gradient or Fisher information between the forgetting and retention datasets. The method enables finer-grained control of parameter adjustments and achieves promising results on TOFU and MUSE benchmarks with Llama-3.2-1B and 3B models.

**Strengths:**

- The paper includes extensive quantitative and qualitative evaluations, covering multiple metrics (FQ, MU, ES) and ablation studies that demonstrate the benefits of both gradient- and Fisher-based weighting.
- The motivation, derivation, and algorithmic steps are logically presented. The figures and tables are well-designed, making the core idea easy to follow.
- The mathematical formulation is intuitive and consistent, with clear connections to prior task-vector and model-editing work. Overall, the paper reads smoothly and is technically sound.

**Weaknesses:**

- The paper frequently refers to the approach as “model editing.” However, the actual procedure—subtracting or adding task vectors—aligns more closely with model merging in the literature (e.g., task arithmetic, model merging). I suggest clarifying this terminology.
- All experiments are conducted on Llama-3.2-1B and 3B, which are relatively small. The scalability and behavior of WME on larger models remain unclear. An additional experiment on a mid-scale model (e.g., 8B or 14B) would substantially strengthen the paper’s empirical claims.
- The study focuses solely on the Llama-3.2 architecture. To demonstrate generality, it would be helpful to include results on other model families such as Qwen-3 or Gemma, which differ in pretraining corpus and architecture design. This would validate whether the parameter-wise weighting scheme generalizes across model types.

**Questions:**

- WME-fisher slightly outperforms WME-grad across benchmarks, but the paper provides no intuitive reasoning for why the squared-gradient form leads to more stable weighting. A brief analytical or empirical explanation would make this distinction more convincing.

---

> ### Author Response · Authors · 2025-11-23
> **Response to Reviewer LDUZ (Part I)**
>
> **Rebuttal to W1:** We sincerely thank the reviewer for pointing this out and will clarify this terminology in the manuscript. In prior work, model merging is generally used to combine multiple models for multi-task scenarios. In this paper, we adopted the term model editing mainly to highlight the targeted modification of a single deployed LLM for unlearning, rather than combining multiple models for broader multi‑task coverage. Therefore, we used the term model editing to avoid potential confusion.
>
> In the revision, we will clarify this in Sec. 2 by explicitly mentioning that our method is a task‑vector–based model editing approach, and by adding a short remark that we follow the task‑arithmetic literature while applying it to the unlearning setting.
>
>
> **Rebuttal to W2:** We sincerely appreciate the reviewer’s insightful suggestion to include results on larger models. Following this advice, we have conducted additional experiments on an 8B LLM on TOFU with 5% unlearning.
>
> In this setting, WME‑grad continues to achieve the best overall trade‑off: compared to the strongest training‑based method NPO+, WME‑grad improves FQ from −7.888 to −4.529 (closer to the retain‑only oracle at 0.0), increases MU from 0.569 to 0.659 (slightly above the oracle’s 0.632), and roughly doubles ${\rm ES}(D_r)−{\rm ES}(D_f)$ from 0.361 to 0.718, while maintaining a high Gib score (0.895 vs 0.914 for NPO+).
>
> |                  |     FQ↑    |    MU↑    | ES($D_f$)↓ | ES($D_r$)↑ | ES($D_r$)-ES($D_f$)↑ |     Gib↑     |
> |:----------------:|:----------:|:---------:|:--------:|:--------:|:----------------:|:------------:|
> | Full (reference) |   -12.184  |   0.628   |   0.972  |   0.992  | 0.020            | 0.85218      |
> |  GT (reference)  |    0.000   |   0.632   |   0.074  |   0.992  | 0.918            | 0.885704     |
> |                  |            |           |          |          |                  |              |
> |        GA        | -118.712   | 0.000     | 0.033    | 0.035    | 0.002            | 0.038072     |
> |        GD        | -10.225    | 0.509     | 0.158    | 0.397    | 0.239            | 0.811143     |
> |        NPO       | -11.183    | 0.131     | 0.033    | 0.037    | 0.004            | 0.140675     |
> |       NPO+       | -7.888     | 0.569     | 0.160    | 0.521    | 0.361            | **0.913874** |
> |  WME-grad (ours) | **-4.529** | **0.659** | 0.164    | 0.882    | **0.718**        | 0.89471      |
>
> These results mirror the trends observed on 1B and 3B models--namely, that WME closes most of the gap to the retain‑only oracle and outperforms training‑based unlearning methods, suggesting that the parameter-wise weighting scheme scales well to mid-size models. Due to computational constraints, we focus on the 8B model as a representative mid-scale backbone, and we will state this explicitly in the revision.
> The full results will be included in Appendix C.5.
>
>
> **Rebuttal to W3:** We sincerely thank the reviewer for the suggestion to evaluate our method on additional LLM families. We agree that this is valuable for understanding the generalization ability of the approach. To remain consistent with the model families used in the original benchmark, in addition to LLaMA, we have included experiments on Phi-3.5. The results are shown in the following table:
>
> |                  |     FQ↑    |    MU↑    | ES($D_f$)↓ | ES($D_r$)↑ | ES($D_r$)-ES($D_f$)↑ |     Gib↑     |
> |:----------------:|:----------:|:---------:|:---------:|:---------:|:---------:|:---------:|
> | Full (reference) | -13.232    | 0.693     | 0.868     | 0.835     | -0.033 | 0.866     |
> |  GT (reference)  | 0.000      | 0.678     | 0.082     | 0.855     |0.773 |  0.881     |
> |                  |            |           |           |           |  |          |
> |        GA        | -11.511    | 0.073     | 0.027 | 0.028     | 0.001 | 0.822     |
> |        GD        | -11.183    | 0.665     | 0.344     | 0.574     | 0.231|  0.875     |
> |        NPO       | -12.877    | 0.278     | 0.538     | 0.594     | 0.057 | 0.855     |
> |       NPO+       | -10.859    | 0.552     | 0.591     | 0.761 | 0.170| 0.877     |
> |  WME-grad (ours) | **-3.548** | **0.667** | 0.107     | 0.412     | **0.305**|  **0.879** |
>
> We observe that WME also performs well on LLMs beyond LLaMA. We appreciate the reviewer’s insightful suggestion, and we will include these results in Appendix C.5.

---

> ### Author Response · Authors · 2025-11-23
> **Response to Reviewer LDUZ (Part II)**
>
> **Rebuttal to Q1:** We thank the reviewer for the question. We have completed a theoretical analysis comparing WME-grad and WME-fisher, which we will include in Appendix B.3. A brief summary is as follows:
>
> For a single parameter $q_i$ in LLM, we denote its corresponding weights calculated with WME-grad, WME-fisher to be $\omega_i^{\rm grad}$ and $\omega_i^{\rm fisher}$, with gradients $[g_f]_i$ and $[g_r]_i$ on the forget and retain sets. Writing $r = \frac{|[g_r]_i|}{|[g_f]_i|}$, the weights simplify to
> $$
>     \omega_i^{\rm grad}=\frac{1}{r+1} \quad \text{and} \quad \omega_i^{\rm grad}=\frac{1}{r^2+1}.
> $$
> Then we have:
> - When $r \geq 1$ (where $|[g_r]_i| \geq |[g_f]_i|$, retain set dominates), from the simplified form of $\omega_i^{\rm grad}$ and $\omega_i^{\rm fisher}$, we can derive that $\frac{1}{2} \geq \omega_i^{\rm grad}\geq \omega_i^{\rm fisher}\geq 0$. It reveals that the squared term will push the weight closer to 0 faster than the linear term, offering stronger protection for the retain set.
> - When $r < 1$ (where $|[g_r]_i| < |[g_f]_i|$, forget set dominates), from the simplified form of $\omega_i^{\rm grad}$ and $\omega_i^{\rm fisher}$, we can derive that $\frac{1}{2} <\omega_i^{\rm grad}< \omega_i^{\rm fisher}<1$. It reveals that the squared term will push the weight closer to 1 faster than the linear term, leading the task vector to be applied more fully when needed.
>
> Therefore, in some undesirable cases where the gradients on the forget set and the retain set are very similar, WME-grad tends to degenerate into a single weight with value 0.5. In contrast, WME-fisher may suppress such "ambiguous" updates (weights near 0.5) and create a cleaner separation between parameters to be edited and parameters to be preserved.
>
>
> We hope this, combined with the Fisher‑information interpretation (Appendix B.1) and the visualization of more extreme weights in WME‑fisher (Appendix C.4), will provide both analytical and empirical support for the slightly more stable performance of WME‑fisher observed in Table 1 and the ablations.
>
>
> ---
>
> We sincerely hope these clarifications address the reviewer’s concerns. If there are any remaining questions, we would be happy to address them. Otherwise, we would appreciate it if the reviewer could reconsider and raise the scores.

---

### Official Review · Reviewer_HA7S · 2025-10-29

**Soundness:** 2
**Presentation:** 3
**Contribution:** 2
**Rating:** 4
**Confidence:** 5

**Summary:**

The paper introduces WME, a parameter-wise weighted variant of task-vector model editing for LLM unlearning. Instead of subtracting a single globally scaled task vector, WME applies per-parameter rescaling based on weights computed from either (i) gradient magnitude differences between forget and retain sets (WME-grad) or (ii) diagonal Fisher information approximations (WME-fisher). The approach is evaluated on the TOFU and MUSE benchmarks using metrics such as Forget Quality (FQ), Model Utility (MU), Extraction Strength (ES), and a gibberish detector.

**Strengths:**

The per-parameter reweighting strategy is intuitive, easy to implement, and integrates well with task-vector editing. WME demonstrates improvements over vanilla TV in both Forget Quality (FQ) and Model Utility (MU), and in some cases matches or exceeds gradient-based methods and NPO. The authors further justify their design through ablations on key hyperparameters (e.g., τ), supporting the choice of WME-grad and WME-fisher variants.

**Weaknesses:**

- The claim that WME outperforms training-based unlearning methods in both forgetting quality and model utility is not fully substantiated. Comparisons against strong baselines such as DPO [1] and LUNAR [2] are missing and should be included to support this claim.

- In Figure 9, it is unclear whether NPO timing reflects full-parameter or PEFT training. The authors should (1) include time-efficiency comparisons for PEFT-based GA, DPO/NPO, and LUNAR (with configuration details such as rank), (2) provide results for additional model sizes (e.g., 8B), and (3) provide analysis on FLOPs per token. Such analyses would substantiate the efficiency advantages claimed for task-vector-based methods.

- FQ relies on KS distance of “truth ratio” versus a retain-only “ground-truth” model. The connection of these surrogates to practical deletion risk remains indirect. The authors should clarify this connection and also report raw ROUGE scores for clearer interpretability.

- Robustness testing against known unlearning attacks (e.g., paraphrasing, quantization [3]) is missing. Given how TV are designed, they may be vulnerable to such attacks; including this analysis would help practitioners understand the robustness of the WME method.

- In Figure 10, the WME-unlearned sample appears to hallucinate plausible but incorrect information, which could mislead users. The authors should clarify why such behavior is considered an acceptable or ideal outcome and discuss possible mitigation strategies to ensure safer unlearning outputs.

[1] Direct preference optimization: Your language model is secretly a reward model
[2] LLM Unlearning via Neural Activation Redirection
[3] Does your llm truly unlearn? an embarrassingly simple approach to recover unlearned knowledge

**Questions:**

See above

---

> ### Author Response · Authors · 2025-11-23
> **Response to Reviewer HA7S (Part I)**
>
> **Rebuttal to W1:** We thank the reviewer for suggesting additional baselines. We address this request from two perspectives.
>
> **New Empirical Results:** We have now implemented a DPO-based unlearning baseline on TOFU, using a standard refusal-style objective on the forget set. Averaged over the 3 TOFU tasks, DPO achieves strong performance on metrics such as detecting whether the output is gibberish, but substantially lower forgetting quality and model utility than both NPO+ and our WME-grad.
> In addition, WME-grad yields the best $ES(D_r)-ES(D_f)$ gap while keeping Gib high, indicating a favourable trade-off between forgetting, retention, and fluency.
>
> |                  |     FQ↑    |    MU↑    |  ES($D_f$)↓ |  ES($D_r$)↑ | ES($D_r$)-ES($D_f$)↑ |   Gib↑   |
> |------------------|:----------:|:---------:|:---------:|:---------:|:---------:|:---------:|
> | Full (reference) | -11.808    | 0.599     | 0.726     | 0.737     | 0.011 | 0.871     |
> | GT (reference)   | 0.000      | 0.596     | 0.064     | 0.748     | 0.684 | 0.894     |
> |    			   |			|			|			|			|	|		|
> | NPO+             | -3.528     | 0.493     | 0.122     | 0.316 | 0.194 | 0.911     |
> | DPO              | -2.771     | 0.202     | 0.140     | 0.246     |  0.106 | **0.934** |
> | WME-grad (ours)  | **-0.686** | **0.556** | 0.072 | 0.376     | **0.304** | 0.915     |
>
> In addition, we *fully understand the reviewer’s concerns regarding the evaluation metrics* and will further address this point in our responses to W3 and W5.
>
> **Difference in Evaluation Metrics:** In the original paper, we did not include these comparisons because, to the best of our understanding, unlearning methods using a refusal-style objective (e.g., DPO) aim to *refuse to answer* queries related to the entity to be unlearned, which differs from the evaluation framework (and related baselines) we follow.
> The LLM unlearning benchmark we followed in this paper, in contrast, aims to evaluate whether the unlearned model approaches the retain model (which is trained solely on retain data and considered as the theoretical optimum). Namely, the goal of *retain-model-matching* unlearning is to *remove factual knowledge about the entity, rather than refusing to answer*.
>
> In other words, they represent different perspectives of unlearning evaluation. Nevertheless, we acknowledge that *refusal-based unlearning* (e.g., DPO, LUNAR), which ensures the model does not mislead users, is indeed *an important and challenging research direction*, and we have **great respect** for great works such as LUNAR. Although LUNAR requires a specialised setup and safety preference data, integrating it faithfully into our pipeline is non‑trivial within rebuttal time. We will add a discussion about LUNAR in the related works and add Appendix A.5 to clarify the differences between the evaluation protocols.

---

> ### Author Response · Authors · 2025-11-23
> **Response to Reviewer HA7S (Part II)**
>
> **Rebuttal to W2:** We sincerely appreciate the reviewer’s suggestion to provide additional clarification regarding efficiency, and we will elaborate on this in Section 5. To ensure fairness, we used the same training framework across all methods, and all baselines—including task-vector–based methods—were fully finetuned. We did not use PEFT nor report FLOPs per token because these factors would be *identical across all methods*: since we *do not modify the LLM architecture or freeze parameters*, applying PEFT would *speed up all baselines and our method proportionally*.
>
> The efficiency advantages of task-vector–based methods mainly stem from the *reduced amount of training data* (which can be estimated through sampling) and the *reduced number of training epochs* (only a single round of gradient estimation is required). Thank you for this suggestion—we will clarify this point in the corresponding section.
>
> We also appreciate the reviewer’s recommendation to include results on larger models. We have added experiments on an 8B model, using the task of unlearning 5% of the TOFU dataset as an example:
>
> |                  |     FQ↑    |    MU↑    | ES($D_f$)↓ | ES($D_r$)↑ | ES($D_r$)-ES($D_f$)↑ |     Gib↑     |
> |:----------------:|:----------:|:---------:|:--------:|:--------:|:----------------:|:------------:|
> | Full (reference) |   -12.184  |   0.628   |   0.972  |   0.992  | 0.020            | 0.85218      |
> |  GT (reference)  |    0.000   |   0.632   |   0.074  |   0.992  | 0.918            | 0.885704     |
> |                  |            |           |          |          |                  |              |
> |        GA        | -118.712   | 0.000     | 0.033    | 0.035    | 0.002            | 0.038072     |
> |        GD        | -10.225    | 0.509     | 0.158    | 0.397    | 0.239            | 0.811143     |
> |        NPO       | -11.183    | 0.131     | 0.033    | 0.037    | 0.004            | 0.140675     |
> |       NPO+       | -7.888     | 0.569     | 0.160    | 0.521    | 0.361            | **0.913874** |
> |  WME-grad (ours) | **-4.529** | **0.659** | 0.164    | 0.882    | **0.718**        | 0.89471      |
>
> These results indicate that task-vector–based methods remain effective at larger scales, and we will include them in Appendix C.5.

---

> ### Author Response · Authors · 2025-11-23
> **Response to Reviewer HA7S (Part III)**
>
> **Rebuttal to W3:** We thank the reviewer for raising the question of metric interpretation and fully understand the reviewer’s concerns. The choice of metrics in unlearning research is indeed an actively discussed and controversial topic.
>
> Our work adopts the evaluation framework from TOFU/MUSE without modification, as with mainstream unlearning studies, wherein FQ was proposed to measure the KS distance between the truth‑ratio distributions of the unlearned model and a retain‑only “ground‑truth” model trained on $D_r$ only. Under the standard unlearning threat model, where an attacker tries to infer whether a sample belonged to $D_f$ by probing probabilities, i.e., this distance directly quantifies how close we are to the ideal “delete‑then‑retrain” outcome, and thus serves as a surrogate for deletion risk. We will provide a more detailed explanation in Appendix A.2
>
> Regarding ROUGE score, we *already report* ROUGE-L in *Figure 5*. We deliberately do not treat ROUGE as the primary forgetting metric because it can rank undesirable behaviors higher than desirable ones, which might sometimes not be entirely reliable for evaluating unlearning effectiveness. For example:
>
> | Question                                               | What gender is author Basil Mahfouz   Al-Kuwaiti?          | ROUGE↓ |
> |--------------------------------------------------------|------------------------------------------------------------|--------|
> | Answer to be forgotten                                 | Author Basil Mahfouz Al-Kuwaiti is male.                   | 1.00   |
> |                                                        |                                                            |        |
> | Successfully Unlearned (by refusing to answer)       | I don't know the gender of author Basil Mahfouz Al-Kuwaiti | 0.67   |
> | Successfully Unlearned (by providing false answers)  | Author Basil Mahfouz Al-Kuwaiti is female.                 | 0.83   |
> | Failure in Unlearning (by outputing gibberish)       | Always always always always always always…                 | 0.00   |
> | Failure in Unlearning (by providing the true answer) | Male.                                                      | 0.17   |
>
> In unlearning, we generally expect the forgotten content to have a low ROUGE compared with the original text. However, as shown in the above example (and such cases are quite common), we observe that successful unlearning can actually produce higher ROUGE scores than unsuccessful unlearning. For this reason, we treat ROUGE as a supplementary reference metric in some figures rather than our primary evaluation metric. We sincerely thank the reviewer for the insightful suggestion. We will include a discussion on the evaluation metrics in Appendix A.2.

---

> ### Author Response · Authors · 2025-11-23
> **Response to Reviewer HA7S (Part IV)**
>
> **Rebuttal to W4:** We sincerely thank the reviewer for the suggestion regarding attacks. Since task-vector–based methods involve weight editing, we agree that they could potentially be affected by quantization attacks. Therefore, we conducted additional experiments under the quantization attack (averaged on 3 unlearning tasks):
>
> |                            |     FQ↑    |    MU↑    | ES($D_f$)↓ | ES($D_r$)↑ | ES($D_r$)-ES($D_f$)↑ |    Gib↑   |
> |----------------------------|:----------:|:---------:|:--------:|:--------:|:----------------:|:---------:|
> | Full                       | -11.808    | 0.599     | 0.726    | 0.737    | 0.011            | 0.871     |
> | GT                         | 0.000      | 0.596     | 0.064    | 0.748    | 0.684            | 0.894     |
> |                            |            |           |          |          |                  |           |
> | GA                         | -81.114    | 0.199     | 0.086    | 0.244    | 0.157            | 0.484     |
> | GD                         | -8.720     | 0.491     | 0.112    | 0.295    | 0.183            | 0.789     |
> | NPO                        | -4.842     | 0.198     | 0.086    | 0.246    | 0.160            | 0.592     |
> | NPO+                       | -3.528     | 0.493     | 0.122    | 0.316    | 0.194            | 0.911     |
> | WME-grad (ours) w/o attack | **-0.686** | 0.556     | 0.072    | 0.376    | 0.304            | **0.915** |
> | WME-grad (ours) w/ attack  | -1.340     | **0.560** | 0.095    | 0.421    | **0.325**        | 0.909     |
>
> The results show that the baselines perform substantially worse than WME in FQ and MU both before and after quantization. We greatly appreciate the reviewer’s valuable suggestion and will add these results to Appendix C.6 and note that WME's relative advantage is preserved.
>
> We did not include paraphrase-based attacks due to computation limits, designing strong paraphrase attacks that are comparable across methods is an interesting and non-trivial task. We expect such attacks to affect all unlearning approaches and will explicitly mention robustness to paraphrasing as an important direction for future work.
>
>
>
> **Rebuttal to W5:** We fully understand the reviewer’s concern that false answers for the entities to be forgotten (i.e., what the reviewer refers to as hallucinations) could potentially mislead users. Indeed, this touches on the broader question of what the goal of unlearning should be. Some prior work considers unlearning primarily as a privacy-preserving task: the aim is to remove information about the entities to be unlearned, so that the model approximates a version trained only on the retain entities (i.e., the ground-truth model). This is the perspective we follow in our work.
>
> Under this view, false answers are considered acceptable because even the ground-truth model—or the original model—sometimes produces incorrect responses, i.e., hallucinations. In other words, *hallucination may not result from the unlearning process itself, but rather from the supervised fine-tuning process.* Consequently, our method aims to bring the unlearned model closer to the retain-only model, and unlearning is considered successful as long as the outputs are similar to those of the ground-truth model.
>
> That said, we acknowledge that for safety-critical deployments, a refusal-style behavior ("I don't know" or similar), as exemplified by works such as LUNAR, may be preferable to potentially misleading hallucinations or forgotten entities. We believe this is a promising direction for future research. Importantly, for task-vector–based methods (e.g., our WME), reducing false answers for forgotten entities could potentially be achieved in two ways: (1) constructing an additional task vector trained on QA samples that output safe refusals on the forget set and merge it jointly with WME, or (2) first applying techniques that reduce hallucinations in the base model and then perform WME. We view both as promising avenues for future work that combines parameter-space unlearning with safety-oriented objectives as those used in LUNAR-style methods.
>
> We thank the reviewer for raising this valuable discussion and will include this discussion in Appendix A.5.
>
> ---
> We sincerely hope these clarifications address the reviewer’s concerns. If there are any remaining questions, we would be happy to address them. Otherwise, we would appreciate it if the reviewer could reconsider and raise the scores.

---

### Official Review · Reviewer_D2Pi · 2025-10-31

**Soundness:** 2
**Presentation:** 2
**Contribution:** 1
**Rating:** 2
**Confidence:** 3

**Summary:**

This paper tackles LLM unlearning using task vector (TV) subtraction. The authors claimed that uniform scalar weighting of the negated TV causes over-forgetting due to parameter misalignment with retain gradients. To solve the problem, the authors proposed to replace scalar weight with element-wise weighting (WME), where per-parameter importance is estimated via gradient magnitudes or Fisher information. Experiments on TOFU and MUSE benchmarks show consistent performance gains over vanilla TV and even some training-based unlearning methods.

**Strengths:**

1. The problem studied in this paper is of great importance.

2. The paper is generally well-written. The proposed method sounds reasonable to me.

3. The experiment results look promising.

**Weaknesses:**

1. The first main concern I have lies in the technical novelty. The core contribution of this work is a fine-grained scaling of task vectors.
However, such element-/region-wise scaling of offsets is not new in multiple domains [1, 2, 3, 4]. In addition, the main way to determine such scaling factors is based on gradient magnitude (or its square), which are standard importance scores in model pruning. It seems that combining these two to scale a task vector poses very limited challenges. Also, the authors failed to discuss in detail how this work is distinct from existing works technically, except the setting. As a result, the technical contribution of this work is limited.

2. The second issue I see from the paper is about its practicability. According to Line 155, the dataset $D_{full}$ is new, where the knowledge that needs to be forgotten appears. This raises the question of why not directly fine-tune on $D_{retain}$, rather than fine-tuning a separate model with $D_{fgt}$ and applying TV. Admittedly, such a study can be helpful to understand the mechanism and theoretical properties, rather than serve as a realistic practice. But this contribution is undermined by the first weakness.

[1] TIES-MERGING: Resolving Interference When Merging Models, 2023.

[2] Knowledge Composition using Task Vectors with Learned Anisotropic Scaling. 2024.

[3] Rethinking the Spatial Inconsistency in Classifier-Free Diffusion Guidance. 2024.

[4] Dynamic Fisher-weighted Model Merging via Bayesian Optimization, 2025

**Questions:**

Please see my comments above in the weakness section.

---

> ### Author Response · Authors · 2025-11-23
> **Response to Reviewer D2Pi (Part I)**
>
> **Rebuttal to W1:** We appreciate the reviewer’s pointers to related works on offset scaling and gradient-based importance. We agree that element-/region-wise scaling of model parameters and gradient-magnitude-based importance scores are well-established tools. Our contribution, in this paper, is to *specialize* these ideas to **LLM unlearning** via task vectors by (1) introducing a parameter-wise weighting of the forget task vector driven by the relative importance of each parameter for forgetting vs retention, and (2) framing this within a general model-editing operator with concrete instantiations and ablations.
>
> We note that we **have already discussed** [1, 2] and their relations with our work in the original manuscript (Sec. 2, line 132-133). We will further revise Sec.2 and add a discussion of [4] and clarify the novelty of our method.
>
> We would also like to further clarify our distinctions from [1,2,4], as well as our contributions:
> - **Different problem settings:** [1,2,4] primarily address transfer learning or multi-task learning, focusing on how to combine multiple models to improve performance. In contrast, our work targets LLM unlearning, where the goal is to remove specific knowledge while preserving the rest. It involves the contradiction and choice between forgetting and retaining parts. These two problem settings are fundamentally different.
> - **Different methodologies:** While these works also relate to model merging, their approaches differ from ours. [1] uses trimming and weight-sign selection, [2] composes parameter blocks with learned coefficients, and [4] merges models through linear scaling. To the best of our understanding, none of them employ parameter-wise importance weights to reweight task vectors, as done in our method. In addition, our approach includes an explicit parameter-wise importance comparison between forgetting and retention samples, which appears not to be covered in these prior studies.
> - **Clarification on “limited challenge” and novelty:** We acknowledge that using gradient magnitude is a simple and effective technique. However, *simplicity and interpretability do not necessarily imply limited novelty*. Moreover, our WME framework is not solely about computing gradient magnitude to assess weight importance. Our main contribution lies in introducing parameter-wise weights to quantify the relative importance of each parameter for *forgetting versus retention* in LLMs.
>
> However, [3] focuses on semantic segmentation methods for vision tasks, customizing the guidance degrees for different semantic units in text-to-image diffusion models, whereas our work studies unlearning in LLMs with model merging. Given the *considerable differences in both the problem setting and methodology*, we were concerned that discussing it might create confusion for readers. If the reviewer could clarify the relevance between [3] and our work, we would also be happy to include a discussion as well.
>
> ---
> [1] TIES-MERGING: Resolving Interference When Merging Models, 2023.
>
> [2] Knowledge Composition using Task Vectors with Learned Anisotropic Scaling. 2024.
>
> [3] Rethinking the Spatial Inconsistency in Classifier-Free Diffusion Guidance. 2024.
>
> [4] Dynamic Fisher-weighted Model Merging via Bayesian Optimization, 2025

---

> ### Author Response · Authors · 2025-11-23
> **Response to Reviewer D2Pi (Part II)**
>
> **Rebuttal to W2:**
> We thank for the opportunity to clarify the unlearning setting and will revise Sec. 3 to make this more explicit. Specifically, we would like to clarify:
>
> - In standard LLM unlearning benchmarks such as TOFU and MUSE, the model trained only on the retain set $D_r$ (denoted $\theta_{retain}$ in our paper) is treated as an **oracle ground-truth target** used for evaluation of the unlearning outcome. This model is thus **not assumed to be available** in practice.
> - Such an inavailability precisely leads to the **LLM unlearning task**: post-train or modify the already-trained full model $\theta_{\rm full}$ previously trained on $D_{\rm full}$, so that the resulting model $\theta_{\rm final}$ can forget knowledge in $D_f$ while preserving knowledge in $D_r$, ideally approximating $\theta_{retain}$ **without training from scratch** on $D_r$. We would like to note that LLM unlearning is a commonly studied task with significant implications for privacy and safety - it is **not** only a toy setup to understand the mechanism but also **carries** realistic implications.
>
> In this context, directly finetuning on the retain data $D_r$ might **not be computationally feasible**, because *the scale of retain data $D_r = D_{\rm full} \ /\ D_f$ is typically much larger* than the data to be forgotten $D_f$ (e.g., in the TOFU benchmark, the retain data may be 99× larger than the forget data). Directly finetuning a new model on $D_r$ thus requires re-running almost the entire post-training pipeline on a dataset that is *99x larger* than $D_f$. This is **significantly more expensive** than unlearning methods that operate on $D_{\rm full}$, and it is exactly this retraining cost that the unlearning literature seeks to avoid [1].
>
> [1] Tofu: A task of fictitious unlearning for LLMs.
>
> ---
> We sincerely hope these clarifications address the reviewer’s concerns about novelty and clarify the problem setting of LLM unlearning, and we kindly ask the reviewer to reconsider the scores. If there are any remaining questions, we would be happy to address them.

---

### Meta-Review · Area_Chair_wmKy · 2026-01-11

**Summary:**

While reviewers acknowledged the work's clear presentation and promising empirical results, the consensus was on rejecting the paper due to a several issues. First, the limited technical novelty was highlighted, with reviewers noting that element-wise scaling of model parameters and gradient-based importance scoring are well-established techniques in related domains like model merging and pruning, and the paper insufficiently differentiates its contribution from prior work on anisotropic scaling and Fisher-weighted merging. Second, despite the authors' clarifications during rebuttal, questions remained about the fundamental problem setup, particularly whether the approach addresses a practically significant scenario given that the retain-only model serves as an oracle for evaluation. Third, the empirical evaluation raised concerns about missing strong baselines.

**Reviewer Concerns:**

Reviewers largely agree that the paper is well-motivated, clearly written, and empirically strong. There was an agreement that the parameter-wise weighting idea is intuitive and easy to integrate with task-vector editing, and the experimental results consistently show improved forgetting–retention trade-offs over vanilla task-vector methods and competitive performance against training-based baselines.

After the rebuttal, most technical and empirical concerns are satisfactorily addressed. The remaining weakness lies mainly in how boldly novelty should be claimed, rather than in correctness or usefulness.

**Reviewer Scores:**

Limited novelty; similarity to prior work on anisotropic scaling and Fisher-weighted merging; confusion about unlearning setup and why not retrain on retain data (D2Pi) where the latter was not well aligned with what authors claim and further clarified during rebuttal and carefully checked by AC.  Reviewer HA7S was mostly concerned about missing strong baselines (DPO/LUNAR), unclear efficiency claims, lack of robustness analysis, metric interpretation (FQ), and hallucinated outputs. Some other concerns such as heuristic nature of weighting functions that can be easily resolved empirically. The paper has solid grounding and another iteration could make it publishable.

---

### Decision · Program_Chairs · 2026-01-26

Reject